# Structural variation of types IV-A1- and IV-A3-mediated CRISPR interference

R. Čepaitė[1,11], N. Klein [2,11], A. Mikšys [1,9,11], S. Camara-Wilpert[3], V. Ragožius [1], F. Benz [4,5], A. Skorupskaitė [1], H. Becker [2], G. Žvejytė [1], N. Steube [6], G.K.A Hochberg [6,7,8], L. Randau [2,8,12], R. Pinilla-Redondo [3,12], L. Malinauskaitė [1,10,12] ✉ & P. Pausch [1,12] ✉

CRISPR-Cas mediated DNA-interference typically relies on sequence-specific binding and nucleolytic degradation of foreign genetic material. Type IV-A CRISPR-Cas systems diverge from this general mechanism, using a nuclease-independent interference pathway to suppress gene expression for gene regulation and plasmid competition. To understand how the type IV-A system associated effector complex achieves this interference, we determine cryo-EM structures of two evolutionarily distinct type IV-A complexes (types IV-A1 and IV-A3) bound to cognate DNA-targets in the presence and absence of the type IV-A signature DinG effector helicase. The structures reveal how the effector complexes recognize the protospacer adjacent motif and target-strand DNA to form an R-loop structure. Additionally, we reveal differences between types IV-A1 and IV-A3 in DNA interactions and structural motifs that allow for in trans recruitment of DinG. Our study provides a detailed view of type IV-A mediated DNA-interference and presents a structural foundation for engineering type IV-A-based genome editing tools.

Clustered Regularly Interspaced Short Palindromic Repeats (CRISPR) and their CRISPR-associated (Cas) proteins are adaptive immune systems of bacteria and archaea that operate against parasitic mobile genetic elements (MGEs), such as phages[1]. Their function centers around short RNAs that guide Cas effector nuclease complexes to previously encountered MGEs for nucleolytic degradation[2]. Although typically present in prokaryotic genomes, recent studies revealed widely distributed CRISPR-Cas systems on MGEs[3–5]. One of the more prevalent CRISPR-Cas systems carried by MGEs are type IV-A systems, which mainly associate with conjugative plasmids to regulate host genes and to interfere with the propagation of competing MGEs[6–8].

Type IV-A are class 1 CRISPR-Cas systems, composed of a multi-subunit effector complex and a CasDinG (Csf4, hereafter referred to as DinG) helicase for transcriptional repression of targeted genes[6,9,10] (Fig. 1a). Type IV-A systems are further subdivided into monophyletic subtypes IV-A1, IV-A2 and IV-A3, based on their Cas protein sequence similarity and Cas8 (Csf1) absence in subtype IV-A2[6]. Additionally, rare type IV-A variants were recently reported and characterized that feature a DinG-fused HNH endonuclease for processive DNA degradation[11]. Type IV systems frequently lack the CRISPR-adaptation proteins Cas1 and Cas2, which are functionally substituted for by host-encoded Cas1 and Cas2 homologs in type IV-A3 systems[9,12]. Processing

[1]LSC-EMBL Partnership Institute for Genome Editing Technologies, Life Sciences Center, Vilnius University, Vilnius, Lithuania. [2]Department of Biology, Philipps-Universität Marburg, Marburg, Germany. [3]Department of Biology, Section of Microbiology, University of Copenhagen, Copenhagen, Denmark. [4]Synthetic Biology, Institut Pasteur, Université Paris Cité, CNRS UMR6047, Paris, France. [5]Microbial Evolutionary Genomics, Institut Pasteur, Université Paris Cité, CNRS UMR3525, Paris, France. [6]Evolutionary Biochemistry Group, Max Planck Institute for Terrestrial Microbiology, Marburg, Germany. [7]Department of Chemistry, Philipps-Universität Marburg, Marburg, Germany. [8]Center for Synthetic Microbiology (SYNMIKRO), Marburg, Germany. [9]Present address: ATEM Structural Discovery GmbH, Remscheid, Germany. [10]Present address: BioNTech UK Ltd, Francis Crick Ave, Cambridge Biomedical Campus, Cambridge, UK. [11]These authors contributed equally: R. Čepaitė, N. Klein, A. Mikšys. [12]These authors jointly supervised this work: L. Randau, R. Pinilla-Redondo, L. Malinauskaitė, P. Pausch. ✉e-mail: lina.malinauskaite@biontech.co.uk; patrick.pausch@gmc.vu.lt

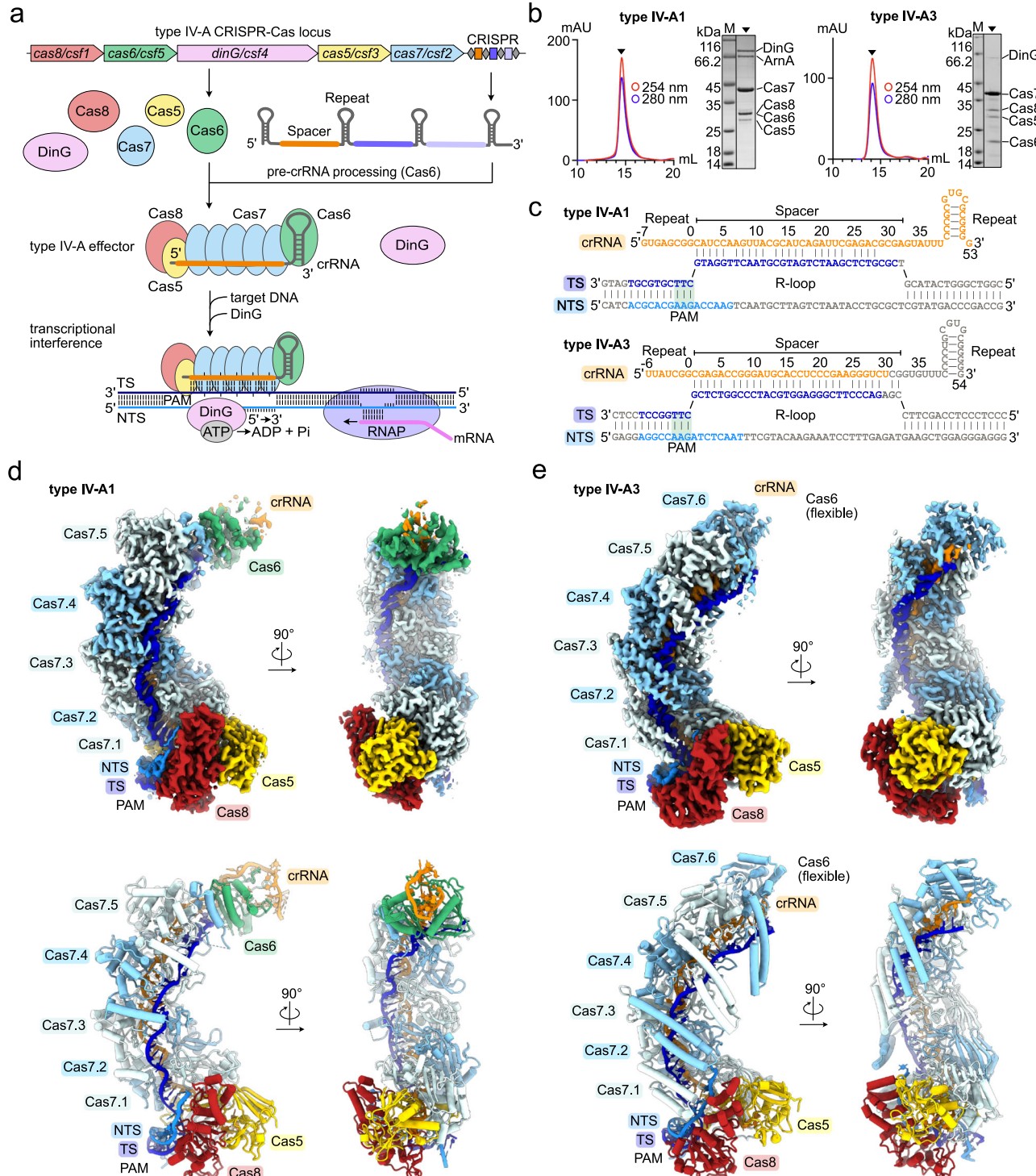

**Fig. 1 | Structures of types IV-A1 and IV-A3. a** Scheme illustrating a CRISPR-Cas type IV-A locus and its transcriptional interference function. Effector subunits are color-coded according to their identity throughout the manuscript. **b** Analytical size-exclusion chromatography traces and SDS-PAGE of purified types IV-A1 (left) and IV-A3 (right) effector complexes. ArnA was identified by mass spectrometry as a purification artifact. *n* = 1. Source data are provided as a Source Data file. **c** crRNA and DNA substrate sequences of types IV-A1 (top) and IV-A3 (below). Bases resolved in the cryo-EM maps (**d**, **e**) are colored orange (crRNA) and blue (DNA). Observed or predicted base pairs are indicated by links between colored or gray bases, respectively. **d** and **e** Sharpened experimental cryo-EM maps (top) in two 90°-rotated orientations of the type IV-A1 (**d**) and type IV-A3 (**e**) effectors in complex with DNA. Unfiltered experimental and EMReady maps are shown in Supplementary Figs. 2–4. Below: structure models of the type IV-A1 (**d**) and type IV-A3 (**e**) effectors in complex with DNA in two 90°-rotated orientations.

of the CRISPR-array-transcribed precursor CRISPR-RNA (pre-crRNA) by Cas6 (Csf5) endoribonucleases yields mature crRNA guides containing individual MGE-derived spacers[13–15], which recruit the Cas proteins Cas5 (Csf3), Cas8 (Csf1) and several Cas7 (Csf2) subunits for formation of a crescent-shaped effector complex[13] (Fig. 1a). To identify a cognate DNA target, the effector complex recognizes a short protospacer adjacent motif (PAM[16]; present upstream of the crRNA-complementary protospacer target), prior to target sequence

interrogation[9,12] (Fig. 1a). After DNA target binding, the ATP-dependent DinG helicase contributes to efficient transcriptional interference in vivo[9,17] (Fig. 1a). This interference activity decreases transcript levels to modulate the host's phenotype, or to reduce the fitness of competing plasmids and phages upon targeting of essential genes[9,12,18]. Notably, CasDinG proteins are distinct from chromosomally encoded non-Cas DinG proteins, suggesting a divergent function[19]. It has been speculated that cellular nucleases may degrade the single stranded DNA produced by DinG after target unwinding[20]; however, the absence of detectable deletions and toxicity when targeting chromosomally encoded genes makes this proposed interference mechanism unlikely[9,12]. Structures revealed that DinG associates with the center of the *Pseudomonas aeruginosa* type IV-A1 transcriptional interference complex at an interface formed by Cas7, when bound to a non-native nicked DNA substrate[20]. While the crRNA biogenesis mechanism by Cas6 and helicase activity of DinG have been well characterized by structure-based studies[13–15], structures elucidating the effector complex beyond the type IV-A1 *P. aeruginosa* complex are lacking.

In contrast to the little studied type IV-A, diverse DNA-targeting type I systems of class 1 CRISPR-Cas have been characterized extensively[21]. Canonical type I CRISPR-associated complexes for antiviral defense (Cascade) adopt sea-horse-like architectures that assemble from: a Cas6 bound to the 3′-end of the crRNA, a helical backbone filament of several subunits of Cas7 that forms along the DNA-target complementary crRNA segment, along with a belly of Cas11 subunits, and a crRNA 5′-end capping Cas5 subunit, which recruits the large subunit Cas8 at the base of the structure[21–28]. To identify a DNA target, Cascade complexes first recognize the PAM motif via Cas8 before hybridizing the DNA target strand (TS) with the crRNA spacer along the Cas7 backbone[21]. Concomitantly, the non-target strand (NTS; complement to the TS) becomes displaced from the TS to form an R-loop structure[21]. R-loop formation results in a conformational shift in the Cascade complex, facilitated by a concerted rearrangement of Cas11 and Cas8[21]. This permits recruitment of the ATP-dependent helicase-nuclease Cas3 at an interface formed by the large subunit Cas8 to engage the single stranded NTS[21]. Cas3 engagement first entails an endonucleolytic cleavage of the NTS, prior to DNA loading into the helicase domain for processive DNA degradation[29]. In the remotely related type IV-A CRISPR-Cas systems, Cas8 homologs are substantially smaller, Cas11 proteins are lacking and any DNA-nuclease activity is absent (Fig. 1a). It is thus unclear how type IV systems recruit and activate the effector helicase DinG for transcriptional interference.

Here, we present several cryo-EM structures of two evolutionary distant type IV-A1 and type IV-A3 complexes (from *Pseudomonas oleovorans* and *Klebsiella pneumoniae*, respectively) bound to cognate DNA targets with and without DinG. The structures reveal multisubunit DNA-surveillance complexes that intimately recognize their DNA-targets to form an R-loop structure. Notably, a bipartite interface formed by Cas8 and Cas5 recognizes the PAM. Upon hybridization of the TS to the crRNA, Cas8 and Cas7 guide the single stranded NTS away from the complexes. Finally, we present the structures of the types IV-A1 and IV-A3 effector complexes bound to DinG: We show how two Cas7 subunits recruit DinG for handover of the NTS in type IV-A1. In contrast, a divergent DinG interface that involves Cas7, Cas8 and Cas5 facilitates DinG recruitment in type IV-A3. Our results support a model in which DNA-binding is achieved analogously to type I Cascade, while recruitment of DinG differs mechanistically, even in closely related subtypes. Together, the structures provide a detailed view of the type IV-A interference pathway and present a structural basis for the engineering of type IV-A-based genome editing tools.

## Results

### Architecture of type IV-A1 and IV-A3 effector complexes
To gain insights into the general architecture and DNA surveillance mechanism of type IV-A systems, we focused on two evolutionary

distant and structurally uncharacterized systems from *P. oleovorans* (type IV-A1)[9] and *K. pneumoniae* (type IV-A3)[12]. Heterologous co-expression in *Escherichia coli* of the type IV-A genes *cas8* (*csf1*), *cas7* (*csf2*), *cas5* (*csf3*), *cas6* (*csf5*) and *dinG* (*csf4*), together with a minimal CRISPR-array (composed of a single repeat-spacer-repeat unit), was followed by a two-step His6-tag affinity and size exclusion purification, revealing the formation of multisubunit ribonucleoprotein (RNP) complexes (Fig. 1b and Supplementary Fig. 1). Cryo-EM structure determination attempts of the binary states in absence of a DNA-target were unsuccessful, likely due to an inherent flexibility of the complexes. To trap the complexes in a more rigid conformation suitable for structural analysis, we determined the DNA-bound ternary state structures. To favor R-loop formation, we complexed purified RNPs with DNA substrates that lacked complementarity between the TS and NTS within the crRNA spacer complementary region (Fig. 1c, Supplementary Fig. 1 and Supplementary Table 1). Cryo-EM structure determination of the resulting complexes revealed the effector complexes at average resolutions of ~3.0 Å (type IV-A1) and ~2.9 Å (type IV-A3) (Supplementary Figs. 2–4).

Both complexes resemble shrimp-like arrangements, with multiple Cas7 subunits forming a backbone along the crRNA spacer (Fig. 1d, e). Five Cas7 subunits assemble a helical filament along the crRNA spine in type IV-A1, while six subunits form the Cas7-backbone in type IV-A3. This facilitates crRNA and TS-DNA hybridization for sequence identification (Fig. 1d, e). We did not observe pronounced cryo-EM density for the single stranded (ss)NTS-DNA beyond the first PAM-proximal nucleotides or the PAM-distal double stranded (ds) DNA, indicating their flexibility (Fig. 1d, e). At the head of the structures, Cas6 caps each complex at the 3′-end of the crRNA (Fig. 1d, e). The low local resolution of the type IV-A3 effector complex around Cas6 indicates flexibility and prevented model building of Cas6, the predicted crRNA hairpin and some segments of the terminal Cas7.6 (Fig. 1e, Supplementary Fig. 5). The base of each complex is formed by Cas5, which associates with the crRNA 5′-end and Cas8 (Fig. 1d, e). Inspection of the cryo-EM density at the 5′-end of the crRNA suggests the presence of a 5′-OH moiety at the terminal nucleotides in positions -7 in type IV-A1 and -6 in type IV-A3 (Supplementary Fig. 6). This is consistent with the Cas6-catalyzed pre-crRNA processing mechanism, which results in 5′-OH and 2′-3′-cyclic phosphate termini, as observed in type IV Cas6 and other Cas6 homologs[13,30]. The observed length of the Cas5-bound crRNA 5′-tag, comprising 8 nucleotides (positions −7 to 0) in type IV-A1 and 7 nucleotides (positions −6 to 0) in type IV-A3, agrees with our previous RNAseq analysis of both complexes[9,12]. Overall, the structures compare well to type I effectors[26,29], the type IV-A1 complex from *P. aeruginosa*[20], and a partial type IV-B structure[31] (Supplementary Fig. 7). Thus, they provide support for a common evolutionary ancestor for type IV and type I systems[10].

### Cas5 and Cas8 form a bipartite PAM recognition interface
The majority of type I Cascade effector complexes initiate DNA sequence interrogation by recognizing the PAM motif via Cas8[21], though type I-F2 employs a divergent Cas5 homolog instead[32]. The *P. aeruginosa* type IV-A1 effector complex also employs Cas8 for PAM recognition[20]. Notably, we found Cas8 and Cas5 interacting with the dsDNA PAM in both complexes: The bipartite interface is formed by a loop extending from the Cas5 thumb motif and the zinc-finger-(Znf)-containing N-terminal domain of Cas8 (Fig. 2a, b).

In type IV-A1, the finger domain of Cas7.1 stabilizes the bipartite PAM interface while stacking against Cas5's thumb motif and the Znf domain of Cas8 (Fig. 2a). Together, Cas8 and Cas5 form a positively charged vice to recruit the 3′-TTN-5′ PAM (ref. 9) and adjacent dsDNA (Supplementary Fig. 8). For PAM sequence recognition, the Cas8 Znf domain probes both the major and minor groove of the dsDNA PAM (Fig. 2c). Cas8 inserts a short α-helix (α3) into the minor groove, with lysine 75 positioned near the A-T PAM base pairs at positions 1 and 2

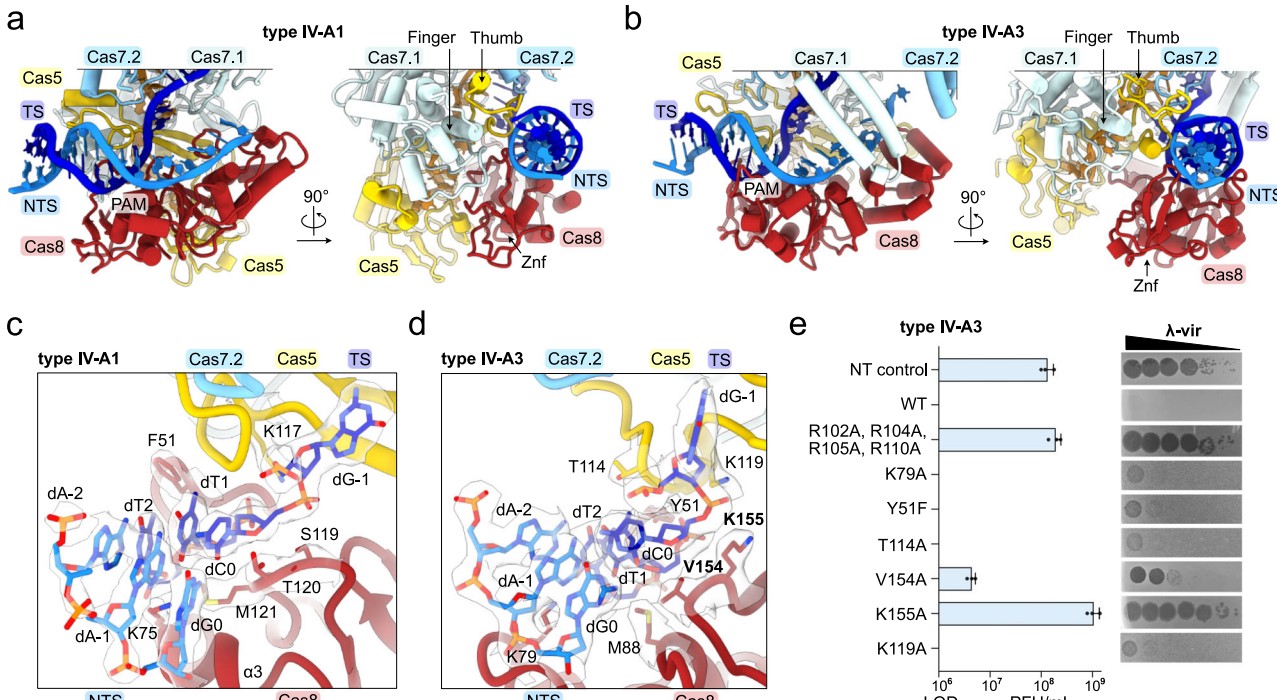

**Fig. 2 | Cas8 and Cas5 mediate PAM recognition. a**, **b** View onto the PAM and adjacent dsDNA bound to types IV-A1 (**a**) and IV-A3 (**b**) in two 90°-rotated orientations. **c** and **d** Close-up view onto the PAM interface in types IV-A1 (**c**) and IV-A3 (**d**). Side chains in proximity to the PAM are shown as sticks. The sharpened experimental cryo-EM map is shown as a translucent surface around the side chains and DNA. **e** λ-vir assay for type IV-A3 probing PAM-interface amino acid substitutions. Plaque Forming Units (PFU) beyond the Limit Of Detection (LOD, countable single plaques) are plotted (blue bars). $n = 3$ independent spot plate replicates; mean ± s.d. Residues producing interference defects upon substitution are highlighted in bold in panel (**d**) and Supplementary Fig. 8b. Source data are provided as a Source Data file.

(Fig. 2c). On the opposite side, phenylalanine 51 aligns to the 5-methyl groups of the thymines in TS positions 1 and 2 of the PAM, stabilized by a short loop extending from the Cas5 thumb (Fig. 2c). Previous structural studies revealed so-called wedge residues that facilitate TS and NTS separation for target identification in various CRISPR effectors[16]. Similarly, type IV-A1 Cas8 inserts a loop to stack the wedge residue threonine 120 against the PAM base pair in position 0 (Fig. 2c). Adjacent to T120, methionine 121 stabilizes the base pair from below (Fig. 2c). To deflect the TS for crRNA-mediated sequence interrogation, serine 119 of Cas8 and lysine 117 of Cas5 clamp the rotated phosphate that connects the PAM and dG1 of the TS (Fig. 2c). Overall, the interactions observed in this type IV-A1 from *P. oleovorans* resemble the mode of PAM interactions employed by the type IV-A1 effector complex of *P. aeruginosa* for a non-canonical 3′-AAG-5′ PAM[20], which is facilitated by a conserved set of amino acid side chains that are important for PAM recognition[20].

Next, we analyzed the mode of PAM interaction in type IV-A3. Like type IV-A1, the dsDNA 3′-TTN-5′ PAM (ref. 12) is recognized by a bipartite interface involving the Znf-containing N-terminal domain of Cas8 and the thumb motif of Cas5 (Fig. 2b). While the finger domain of Cas7 stabilizes both Cas5 and Cas8 in type IV-A1, only the loop extending from Cas5's thumb is contacted by Cas7 in type IV-A3 (Fig. 2b). Although the functional consequences are unclear, the lack of interactions may confer a higher degree of conformational flexibility. Similar to type IV-A1, extensive polar interactions involving Cas5 and Cas8 recruit the PAM-containing dsDNA duplex (Supplementary Fig. 8). However, in type IV-A3, a more pronounced arginine rich loop (RRL; arginines 102, 104, 105 and 110), extending from the Cas5 thumb, inserts into the minor groove of the PAM-adjacent dsDNA (Supplementary Fig. 8). For PAM sequence recognition, Cas8 aligns a loop emanating from its Znf to the minor groove to identify the PAM base pair in position 2, with lysine 79 probing dT2 of the TS (Fig. 2d). On the

opposite side of the PAM, threonine 114 of Cas5 and tyrosine 51 of Cas8 align to the 5-methyl groups of both thymines in TS position 1 and 2 via the major groove (Fig. 2d). Y51 of Cas8 additionally forms a hydrogen bond (distance of ~2.4 Å) between the Y51 hydroxyl group and the dT1 OP1 moiety, directly interacting with the phosphate bond that connects target strand PAM nucleotides dC0 and dT1. While type IV-A1 stacks a wedge residue onto the PAM base pair in position 0, type IV-A3 Cas8 positions valine 154 and methionine 88 in the minor groove adjacent to the base pair in position 0 (Fig. 2b, d). Lysine 155 of Cas8 and lysine 119 of Cas5 clamp in place the rotated phosphate that connects the PAM and dG1 of the TS, which together bend the TS for sequence interrogation (Fig. 2d). In summary, while types IV-A1 and IV-A3 employ comparable modes of PAM interaction, distinct overall conformations and amino acid identities potentially contribute to altered DNA-binding kinetics.

We have previously shown that type IV-A3 interferes with the propagation of phage λ-vir when targeting essential genes[12]. To understand how PAM-interface interactions contribute to interference, we employed our phage-targeting assay to evaluate amino acid substitutions (Fig. 2e). Wild-type (WT) interference protects bacterial cells from phage infection entirely (Fig. 2e), but substituting a set of of arginines in Cas5's thumb motif annulled this protection (R102A, R104A, R105A and R110A; they otherwise form extensive polar contacts with the dsDNA phosphate backbone) (Fig. 2e and Supplementary Fig. 8). This shows that DNA recruitment via Cas5's thumb is critical for interference. We did not observe a loss in protection when we introduced substitutions disrupting polar interactions by the PAM-interacting residues K79A and Y51F of Cas8, or residue T114A of Cas5 (Fig. 2e). Substitution of the wedge residue V154 to alanine resulted in a moderate loss of protection (Fig. 2e), suggesting that strand separation contributes to efficient interference. We next probed K155 of Cas8 and K119 of Cas5, which together pinch the rotated phosphate bond

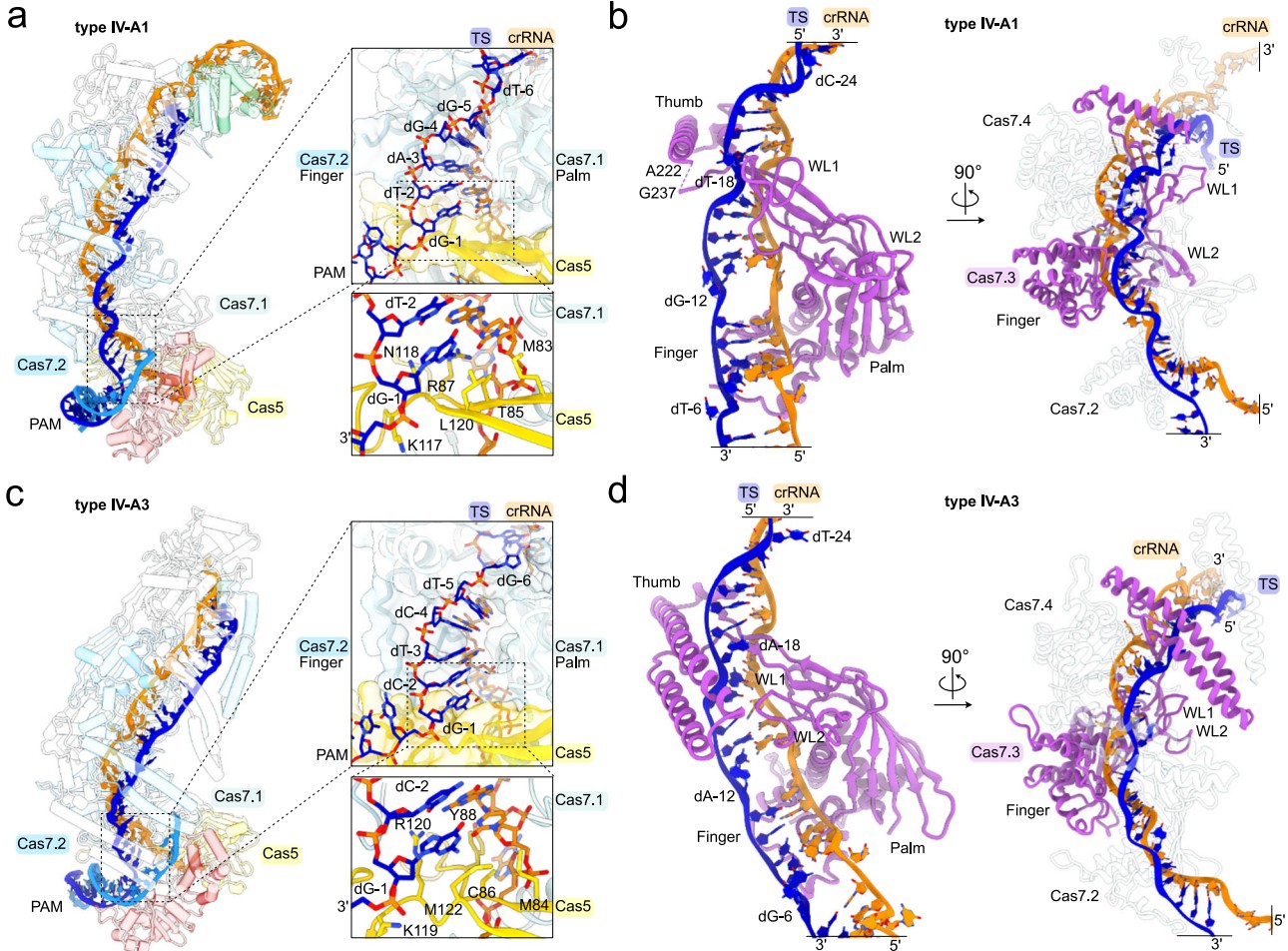

**Fig. 3 | Cas5 and Cas7 stabilize the crRNA:TS hybrid for target interrogation. a, c** Left: overview of the type IV-A1 (**a**) and type IV-A3 effectors (**c**). Cas proteins are shown as translucent cartoons. Right: close-up view on the seed region in TS position −1 to −5 in types IV-A1 (**a**) and IV-A3 (**c**). Notch motif side chains in proximity to the base pair in position −1 are shown as sticks. **b, d** Cas7.3 (purple) of types IV-A1 (**b**) and IV-A3 (**d**) associated with the TS (blue) and crRNA (orange) hybrid in two 90° rotated views. The right panels show the adjacent Cas7 subunits (transparent cartoon).

connecting the target strand PAM and protospacer. Alanine substitution revealed a pronounced loss of phage protection for K155A, while K119A protected against λ-vir at a level similar to the WT system (Fig. 2e). This shows that deflection of the TS by Cas8 is critical for type IV-A3 function, while Cas5 might only play a minor role. Taken together, the data suggest that individual sequence-specific PAM interactions only marginally contribute to efficient interference in vivo, in contrast to interactions that recruit and rearrange the target DNA for R-loop formation.

## Cas7 facilitates target recognition

Subsequent to PAM recognition, DNA-targeting CRISPR-Cas effector complexes generally unwind the dsDNA target for crRNA-mediated sequence interrogation by an ATP-independent mechanism. For sequence interrogation, R-loop formation then proceeds via a directional dsDNA unzipping mechanism that entails first the formation of a so-called PAM-proximal seed[33], followed by full TS-DNA hybridization to the crRNA spacer, as observed for diverse CRISPR effectors, such as Cas9[34–37], Cas12[38,39], type I Cascades[40–42] and the *P. aeruginosa* type IV-A1 effector[20]. To understand how type IV-A complexes facilitate seed stabilization, crRNA:TS hybridization and completion of the R-loop structure, we analyzed the protein:DNA:RNA and protein:protein interfaces along their Cas7-backbones.

Inspection of the PAM-proximal crRNA:TS heteroduplex revealed that Cas5, Cas7.1 and Cas7.2 facilitate initial sequence recognition and seed stabilization in both complexes (Fig. 3a, c). The rerouted TS nucleotide in position −1 is paired up with the first crRNA nucleotide and held in place by a notch in Cas5's thumb domain, which traverses the heteroduplex (Fig. 3a, c). Adjacent to this thumb, the palm domain of Cas7.1 and the finger domain of Cas7.2 sandwich the crRNA:TS base pairs in position −2 to −5, from the belly- and back-side of the complex, respectively (Fig. 3a, c). Similar to other class I CRISPR effector complexes[21], Cas7's thumb deflects every sixth TS nucleotide, preventing pairing with the complementary crRNA nucleotide (Fig. 3b, d).

Along the crRNA:TS heteroduplex, Cas7 assumes an interlocked filament configuration to facilitate crRNA-mediated sequence recognition (Fig. 3b, d). The formation of the Cas7 filament involves aligning adjacent Cas7 protomers through interactions between their palm domains and wrist loops on the belly-side of the complex (Fig. 3b, d). On the back-side of the complex, the thumb domain stacks in between the finger domains of the neighboring Cas7 subunits and folds back over the crRNA:TS heteroduplex, potentially contributing to the stabilization of the RNA:DNA hybrid (Fig. 3b, d).

At the head of the type IV-A1 effector complex, Cas6 is recruited through its N-terminal RRM1 to the thumb of Cas7.4 (Supplementary Fig. 9). Remarkably, the recruitment of Cas6 in this position caps the crRNA:TS hybrid by aligning the RRM domains via threonine 224 and histidine 225 against the terminal base pair formed between dG-28 (TS) and C28 (crRNA) (Supplementary Fig. 9). The terminal crRNA-complementary TS nucleotides are, in consequence, not paired up to

the crRNA and are instead guided away from the complex through a cleft formed by the C-terminal RRM2 of Cas6 and WL1 of Cas7.5 (Supplementary Fig. 9).

Overall, the mechanism of sequence recognition during R-loop formation is comparable to that of type I Cascade effector complexes and the recently described type IV-A1 effector from *P. aeruginosa*[20,21]. However, why type IV-A1 effector complexes do not fully utilize the sequence information of their crRNA is unclear.

## Mismatch tolerance of type IV-A

CRISPR effectors typically tolerate some degree of mismatches between the crRNA and TS. While this plays an important immunological role by providing resilience against rapidly evolving MGEs, it can be disadvantageous for genome editing applications, where off-target effects can lead to undesired outcomes[43]. To evaluate the fidelity of type IV-A1, we set up an efficiency of transformation (EOT) mismatch-tolerance assay in *E. coli* BL21-AI expressing type IV-A1 *cas* genes and crRNA from separate plasmids (Fig. 4a–d). Transformation of a target plasmid reduced the efficiency of transformation only in the presence of a matching PAM sequence and crRNA complementary protospacer sequence, indicative of plasmid interference (Fig. 4c, d). We next mutated the crRNA-encoding plasmid across individual positions of the crRNA (positions 1–16) by changing G to C and A to T, and vice versa. Strains containing either crRNAs C1G or C4G showed higher EOTs and thus reduced plasmid interference compared to the fully matching crRNA. Combining mutations at positions 2–4 raised EOTs to levels comparable to the non-targeting (NT) control. Single mismatches in all other positions 1–16 were well tolerated (Fig. 4c). Next, we generated mutations in the targeted region (protospacer) of the transformed plasmid (Fig. 4d). Here, only mismatches in position 1, 4 and the combination of 2, 3 and 4 resulted in increased EOTs (Fig. 4d). Together, this shows that type IV-A1 interference can tolerate single base pair mismatches along the central PAM-proximal half of the crRNA and that mismatches in the immediate seed region appear less well tolerated. We next introduced quadruple mutations in the PAM-distal segment of the crRNA to evaluate the mismatch tolerance in this region (bases 17–20, 21–24, 25–28 or 29–32) (Fig. 4c). Mismatches in base pairs 17–20 and 25–28 increased EOT to levels comparable to the NT control, while mismatches in base pairs 21–24 and 29–32 lowered EOTs to levels comparable to the no mismatch crRNA. This can be explained by our type IV-A1 structure, which revealed that crRNA nucleotides 29–32 are not paired to the TS (Fig. 4a). Mismatches in position 21–24 might be tolerated as they allow for completion of the R-loop structure, while also licensing DinG recruitment and thus interference.

We next performed an electrophoretic mobility shift assay (EMSA) to directly test the ability of the type IV-A1 complex to bind PAM-distally mismatched DNA in the absence of DinG (Fig. 4e). While DNA mismatched in base pairs 19–23 and 25–28 bound to the complex at a level comparable to the no mismatch DNA, mismatches in base pairs 13–17, 17–20 and 21–24 resulted in a decreased binding ability. Interestingly, our in vivo assay revealed that mismatches in base pairs 17–20 and 25–28 are not tolerated for interference (Fig. 4c), even though the mismatched DNA bound in vitro at levels comparable to mismatches in base pairs 21–24 and the no mismatch control, respectively (Fig. 4e). This suggests that while excessive PAM-distal mismatches only marginally affect R-loop formation, they may impair the downstream interference regulation.

Given the structural differences in the Cas7 thumb domain of type IV-A3, which folds over the crRNA:TS hybrid (Fig. 3b, d), we wondered whether mismatch tolerance differs between types IV-A3 and IV-A1. Probing mismatches for type IV-A3 in base pairs 13–17, 19–23, 25–29 and 31–32 by EMSA revealed strong binding defects for mismatched DNA in base pairs 13–17, 19–23 and 25–29 (Fig. 4f). This is in stark contrast to the mismatch fidelity observed for type IV-A1 (Fig. 4e), suggesting that the DNA-interaction modes differ between the two systems.

To assess the seed mismatch tolerance of type IV-A1 in its native host, we set up a *gfp* mismatch CRISPRi assay in wild-type *P. oleovorans* (Fig. 4g). We observed CRISPRi activity for the no mismatch crRNA *gfp*-guide under basal crRNA expression conditions (Fig. 4h). Individual seed region mismatches (position 1–5) alleviated CRISPRi activity to levels comparable to the NT control (Fig. 4h). In contrast, a mismatch in position 6 did not affect activity (Fig. 4h), which is consistent with an unpaired base 6 of the crRNA (Fig. 4a). This result aligns with in vitro observations of the *P. aeruginosa* type IV-A1 complex, which revealed binding defects for seed-mismatched DNA[20]. Upon IPTG-induced crRNA overexpression, only individual mismatches in base pairs 1–3 reduced CRISPRi activity (Fig. 4h). This might be attributed to increased concentrations of *gfp*-crRNA-bound effector complexes, which may compensate for the attenuated affinity of mismatched crRNA guides.

In summary, the data shows that the type IV-A systems sense mismatches in the seed region and extensive mismatches in the PAM-distal region, demonstrating their function as a robust immune system and informing future genome editing applications.

## Cas8 and Cas7 guide the NTS by divergent mechanisms in types IV-A1 and IV-A3

During R-loop formation, DNA-targeting type I effector complexes either guide the NTS via Cas8 towards a nuclease present within the complex[26], or display the NTS for recruitment of an in-trans acting effector nuclease[29]. Similarly, the *P. aeruginosa* type IV-A1 complex directs the NTS towards DinG, along Cas8's C-terminal domain (CTD)[20].

In our *P. oleovorans* type IV-A1 structure, we could trace the single stranded NTS from the PAM to nucleotide 6 (Fig. 1c, d), while the remaining NTS nucleotides were not visible, likely due to their flexibility. Like in the *P. aeruginosa* complex[20], the NTS is guided along Cas8 in a positively charged trench that traverses the central beta sheet and CTD towards the center of the complex (Fig. 5a). Similar to type IV-A1, the structure of type IV-A3 revealed a cryo-EM density corresponding to the NTS (through nucleotide position 8) atop the C-terminal domain of Cas8 (Fig. 1c, e). The interactions of Cas8's CTD with the NTS accordingly mirrored those of type IV-A1 (Fig. 5a, b). Different from type IV-A1, we found α-helices 8 and 9, originating from the thumb domain of Cas7.1, extending from the back of the type IV-A3 effector complex to interlock with the CTD of Cas8 (Fig. 5b, c). This NTS-interaction mode fully encloses the NTS in a positively charged channel (Fig. 5b, c). Successive truncation of Cas7's thumb domain by substituting segments in α-helices α8 and α9 with a GSSG linker revealed DNA binding defects in an EMSA experiment (Fig. 5d). Analytical size-exclusion chromatography confirmed that the variant complexes are properly folded and soluble (Supplementary Fig. 12). This suggests that the thumb helices contribute to efficient DNA binding.

To explore the conformational heterogeneity of DNA-bound effector complexes, we performed a three-dimensional variability analysis (3DVA) on our datasets. 3DVA revealed a dynamic type IV-A1 effector which may undergo longitudinal contraction (-7 Å) along the backbone upon DNA binding (Fig. 5e). This agrees with a previous observation, showing that the *P. aeruginosa* type IV-A1 effector contracts -10 Å upon DNA binding[20]. However, we did not observe density that would suggest stabilization of the NTS beyond the Cas8 CTD interaction. 3DVA of our type IV-A3 dataset showed a highly dynamic region around the terminal Cas7.6 and Cas6. The analysis further revealed stabilization of the NTS upon association to the Cas7 thumb helices α8, α9 and wrist loops along the Cas7 backbone up to Cas7.3 (Fig. 5f).

Although the functional consequences of the two different NTS-routing mechanisms are unclear, the DNA-channeling mode in type IV-A3 could lead to a tightly bound DNA, as suggested by EMSAs (Fig. 5d). This, in turn, might stabilize the complex on its target DNA in vivo, potentially preventing the removal by MGE-encoded helicases, which

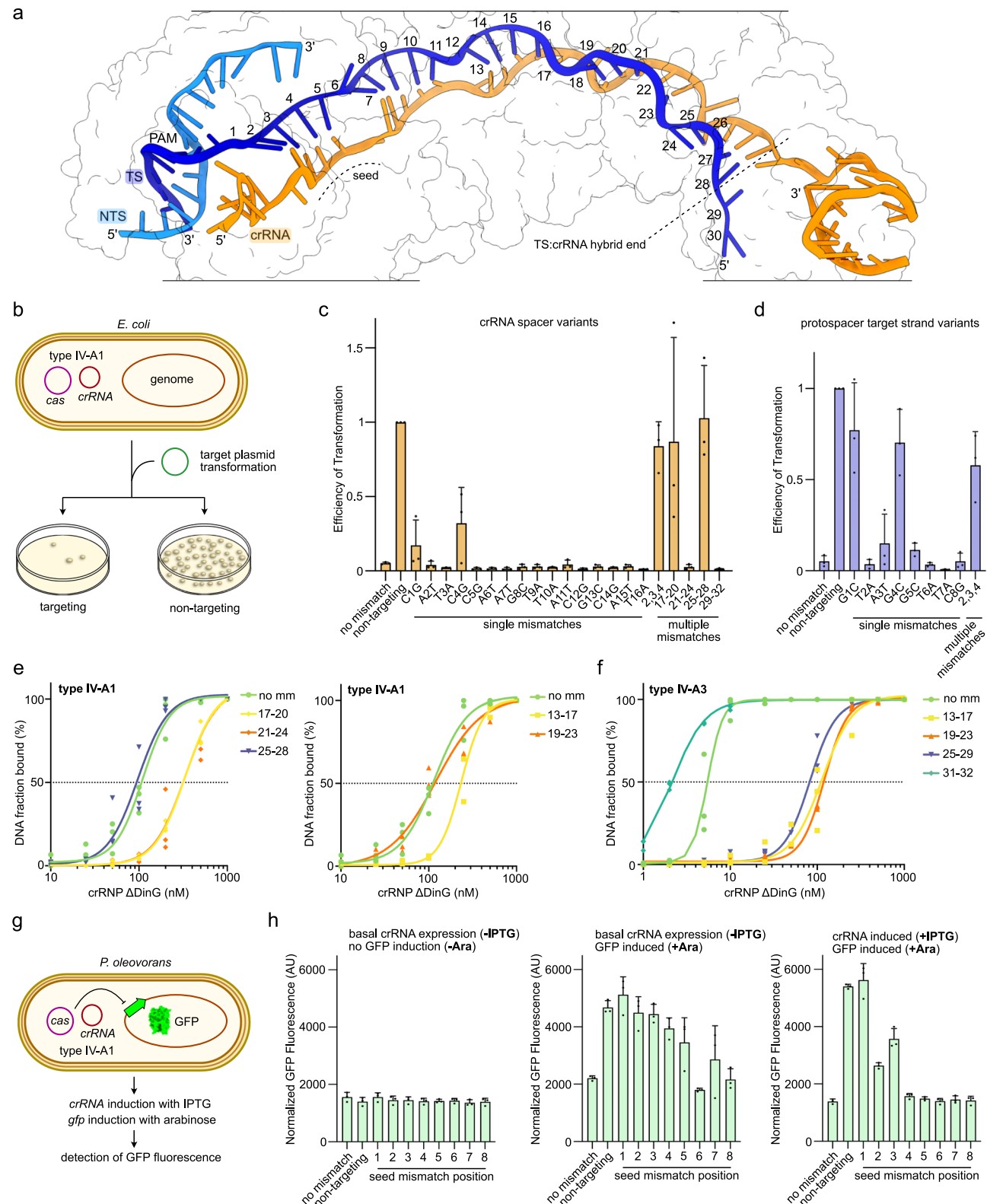

may include DinG itself. In type I systems, anti-CRISPR proteins (Acrs) primarily bind DNA-interaction sites, thereby preventing target binding[44]. Structural variation of the DNA-interaction sites that obscure Acr binding sites may also account for the extended parallel helix motif observed in the Cas7 thumb domain (α8 and α9), analogous to findings for CRISPR-Cas subtype I-F2[32]. Whether the differential NTS routing is involved in helicase or Acr protection requires further investigation.

## Structure of the IV-A1 effector complex in presence of DinG

During heterologous expression and purification, we observed co-elution of DinG with the type IV-A1 complexes, and weaker co-elution with the type IV-A3 complexes (Fig. 1b and Supplementary Fig. 1). To understand if DinG directly interacts with the DNA surveillance complex in absence of a DNA target, we analyzed the size-exclusion eluates by mass photometry (Supplementary Fig. 13). This revealed molecular species corresponding to monomeric DinG and the crRNP DNA

**Fig. 4 | Mismatch tolerance of type IV-A. a** Cryo-EM structure of the *P. oleovorans* type IV-A1 effector complex illustrating the R-loop formed by the target DNA (blue) and crRNA (orange). Protein subunits are shown as outlines. Protospacer target strand nucleotides are labeled according to their position. **b** Scheme illustrating the Efficiency of Transformation (EOT) assay. Cas proteins and crRNA are produced from separate expression plasmids. **c, d** EOT mismatch-tolerance assay with variable mismatching in spacer bases (**c**, orange bars) and protospacer bases on the target strand (**d**, blue bars). *n* = 3 independent replicates; mean ± s.d. Source data are provided as a Source Data file. **e, f** Electrophoretic mobility shift assays testing for the ability of types IV-A1 (**e**) and IV-A3 (**f**) complexes to bind non-mismatched (no mm) and mismatched DNA targets (mismatch ranges are labeled). *n* = 3 independent replicates; the no mm data in panel (**e**) is duplicated and shown in both graphs for reference. EMSA gels are shown in Supplementary Figs. 10, 11. Source data are provided as a Source Data file. **g** Scheme illustrating the *P. oleovorans gfp* repression assay. Type IV-A1 is natively expressed from the *P. oleovorans* megaplasmid and the *gfp*-targeting crRNA is expressed from a separate plasmid. **h** *gfp* repression mismatch tolerance (green bars) in uninduced cells (left), in cells with induced *gfp* (middle), and with induction of both, *gfp* and the *gfp*-targeting crRNA (right). *n* = 3 independent replicates; mean ± s.d. Source data are provided as a Source Data file.

surveillance complex (in absence of DNA) for types IV-A1 and IV-A3 (Supplementary Fig. 13). Upon addition of the DNA-target, we observed complexes corresponding to the theoretical molecular weight of DNA and DinG-bound effector complexes (Supplementary Fig. 13). This suggests that DinG associates with the complex in trans after DNA binding, reminiscent of the Cas3 recruitment mechanism in diverse type I CRISPR-Cas systems.

Inspection of the cryo-EM 2D-classes for the type IV-A1 DNA-bound effector complex revealed an additional density on its belly side (Supplementary Figs. 2, 3). Structure reconstruction revealed DinG at resolutions ranging from ~3.5 Å to ~7 Å, which could be unambiguously assigned to the helicase domains (HD1 and HD2), as well as the FeS-like and arch domains (Fig. 6a, b and Supplementary Fig. 2). The structure further revealed the complete NTS path from the PAM to the exit of the helicase (Fig. 6a, b). We did not observe the PAM-distal single-stranded NTS and TS:NTS duplex or the predicted DNA-binding accessory N-terminal domain (NTD) of DinG[15], indicating a high degree of flexibility for these elements. Mass spectrometry confirmed the presence of the NTD of DinG, suggesting that the domain is flexible, rather than being subject to degradation (Supplementary Fig. 14).

Examination of the DinG interface revealed interactions involving its α-helices 19, 21, 27 and 28 and the wrist loops (WL2) of Cas7.1 and Cas7.2, along with a loop extending from the palm domain of Cas7.1 (Fig. 6b, c). This conformation places the arch domain near the center of the complex and aligns the HD2 of DinG for receiving the NTS, which extends from Cas8's CTD across WL2 of Cas7.1 towards DinG (Fig. 6a, b). At the center of the interface, tryptophan 494 and isoleucine 657 of DinG form non-polar interactions with isoleucine 165 of the palm-loop in Cas7.1 and valine 56 of WL2 in Cas7.2 (Fig. 6c). This non-polar interface core is further surrounded by polar side chains, which might participate in the Cas7-DinG interface (Fig. 6c). 3DVA of the DinG-bound effector showed that the helicase domains, arch and vFeS are highly dynamic and rearrange relative to each other (Supplementary Fig. 15).

To dissect the Cas7.1-DinG interface, we set up a CRISPRi assay in *E. coli* BL21-AI, where *lacZ* repression reports on the effects of interface mutants via blue-white colony screening. In the presence of wild-type type IV-A1 targeting the 5′-region of *lacZ*, blue-white colony ratios indicated strong CRISPRi activity (Fig. 6d). To avoid indirect effects on Cas7's ability to bind DNA, we only introduced individual substitutions in DinG (R440A, W494A K661A, R493A, K63A, I657W or R653A). Only arch domain R440A, HD1 domain W494A and HD2 domain R653A substitutions caused loss of CRISPRi activity (Fig. 6d). Analytical size-exclusion chromatography confirmed that the R440A, W494A and R653A DinG mutants are properly folded and soluble (Supplementary Fig. 16a). This suggests that the interface formed by the HD1, HD2 and arch domains is crucial for DinG recruitment and subsequent transcriptional interference.

Next, we compared our structure of the type IV-A1 effector in complex with DinG with published structures of *P. aeruginosa* type IV-A1[20]. Superimposition of our structure with the proposed *P. aeruginosa* DinG effector recruitment state (state I) revealed nearly identical overall conformations (root-mean-square deviation RMSD of 1.8 Å; Supplementary Fig. 17). However, we observed a slight repositioning of

WL2 in Cas7.1 of ~4 Å and a minor offset between the helicase domains of DinG in our structure (Supplementary Fig. 17). Trapping *P. aeruginosa* DinG in the presence of a non-native nicked DNA substrate revealed two distinct states with DinG bound to the effector: either at Cas7.1 and Cas7.2, or at Cas7.2 and Cas7.3[20]. Based on this observation, the authors proposed a sliding mechanism along the Cas7 backbone[20]. The conformation observed in our structure when complexed with a native DNA target might represent the initial DinG recruitment state, prior to the sliding, or pre-sliding state.

## Structure of the DinG-bound IV-A3 effector complex

Our initial type IV-A3 cryo-EM dataset had too few particles of the type IV-A3 effector complex in presence of DinG (Supplementary Fig. 3), despite mass photometric evidence for complex formation (Supplementary Fig. 13). Cryo-EM grid optimization and exhaustive data collection yielded a sufficient amount of particles to reconstruct three structures of the DinG-bound type IV-A3 effector complex, with DinG resolved at ~3 Å to ~7 Å in three distinct states (states I–III) (Fig. 7a, b, Supplementary Table 2 and Supplementary Figs. 18, 19). However, density was lacking for the arch domain in state II and only present at low resolution in state III (Fig. 7b). The structures also revealed that type IV-A3 DinG features no additional domains, such as the accessory N-terminal domain of type IV-A1 DinG.

Similar to the type IV-A1 structure (Fig. 6), type IV-A3 DinG is recruited to the NTS at the belly side of the complex (Fig. 7a). But unlike type IV-A1 DinG, type IV-A3 DinG contacts the palm domains of Cas5, Cas7.1 and Cas7.2 through HD2 domain α-helices 22–25, and interacts with the thumb-domain α-helices 8 and 9 of Cas7.3 (Fig. 8a). Moreover, arch domain α-helices 14 and 15 are near Cas7's WL motif, and DinG inserts its C-terminal tail into a cleft formed between Cas8's CTD and the palm of Cas5, thereby anchoring the HD2-domain poised for NTS recruitment (Fig. 8a). Superimposition of types IV-A1 and IV-A3 DinG revealed that type IV-A1 DinG lacks the elements that facilitate interactions between type IV-A3 DinG and Cas8's CTD as well as the palm domains of Cas5, Cas7.1 and Cas7.2 (Supplementary Fig. 20), underlining the presence of different DinG recruitment interfaces across these systems.

To dissect the type IV-A3 DinG interface, we tested amino acid substitutions in DinG using our lacZ interference assay (Fig. 8b). Substitutions at the DinG HD2 domain interfaces with Cas5 palm (R605A/R608A, L588A, L588W) and Cas7.3 α-helices 8 and 9 (N412A/I418A/Q422A) did not disrupt interference activity, as evidenced by the absence of blue colonies (Fig. 8b). A mutation at the DinG arch with Cas7.4 WL interface (K288A) did not result in blue colonies, either (Fig. 8b). Deletion of the Cas8/Cas5-interacting C-terminal tail of DinG (residues 617-624) caused a slight reduction of interference (Fig. 8b). In contrast, substitutions at the DinG HD2 to Cas7.1 and Cas7.2 palm interfaces (Y527A and R537A/T538W/F539A) abrogated interference, similarly to the non-targeting guide (Fig. 8b). Analytical size-exclusion chromatography confirmed that the Y527A and R537A/T538W/F539A DinG mutants are properly folded and soluble (Supplementary Fig. 16b). We also substituted tyrosine 527, which extends into the interface core from α-helix 22, by a tryptophan to assess whether non-polar contacts facilitate the interaction. The Y527W substitution did

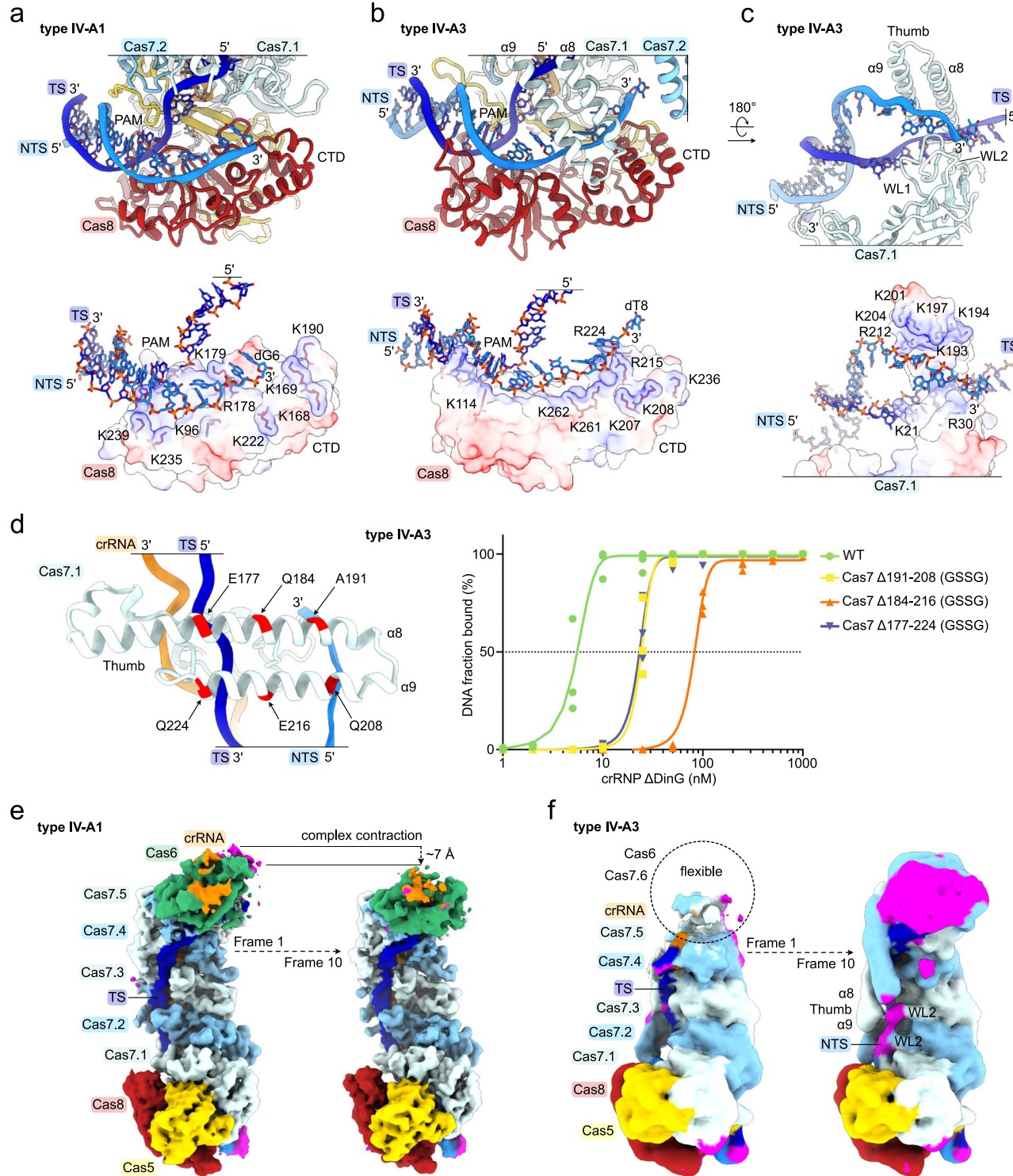

**Fig. 5 | Cas8 and Cas7 guide the NTS. a**, **b** Close-up view on the NTS-trench in the type IV-A1 (**a**) and type IV-A3 (**b**) effectors. Top: cartoon model; below: charge surface representation of Cas8. The target and non-target DNA strands, as well as arginine and lysine residues of Cas8 in proximity of the DNA are shown as sticks. **c** Close-up view on Cas7. Cas8 is hidden for clarity. Top: cartoon model; below charge surface representation. Structural elements and side chains in proximity of the DNA are labeled. **d** Right: electrophoretic mobility shift assay testing for the ability of type IV-A3 complex Cas7 variants to bind DNA targets. Left: scheme illustrating the GSSG-substitution positions in Cas7. $n = 3$ independent replicates; the *WT* data is duplicated for reference from Fig. 4, panel E (no mm). EMSA gels are shown in Supplementary Fig. 12. Source data are provided as a Source Data file. **e**, **f** Two frames of the 3DVA (left: frame 1; right: frame 10) for the DNA-bound type IV-A1 (**e**) and type IV-A3 (**f**) effectors. Magenta colored map regions indicate densities not accounted for by our models.

not cause an interference defect (Fig. 8b), suggesting that non-polar contacts primarily facilitate the interaction. Collectively, these results suggest that the interface between DinG HD2 α-helices (α22, α23) and Cas7's palm primarily determines DinG recruitment.

The lacZ interference assay also revealed that the interference activity of type IV-A3 is stronger than the interference activity of type IV-A1 (Figs. 6d and 8b). This observation agrees with our EMSA results, which revealed approximately ten-fold tighter binding of the target

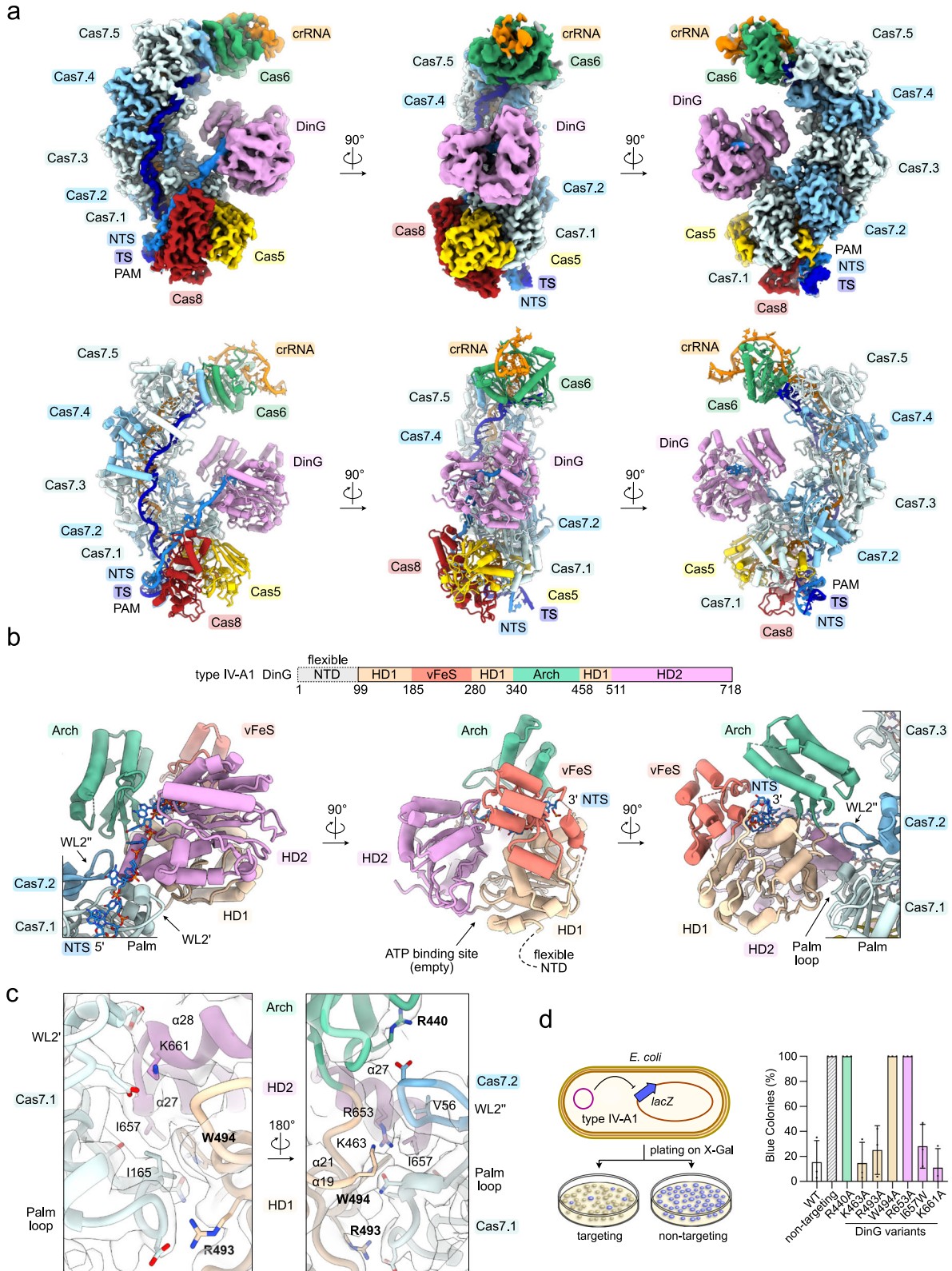

**Fig. 6 | Cryo-EM structure of type IV-A1 in complex with DinG. a** Top: Sharpened experimental cryo-EM map of the type IV-A1 effector in complex with DinG. Unfiltered and EMReady maps are shown in Supplementary Figs. 2, 4. Below: Structure model of the type IV-A1 effector in complex with DinG. **b** Top: domain organization scheme of DinG. Below: DinG-centered views. DinG domains are color coded according to the scheme above. **c** Close-up view onto the Cas7-DinG interface in two 180°-rotated orientations. The sharpened experimental cryo-EM map is shown as a translucent surface. **d** *lacZ*-CRISPRi assay probing amino acid substitutions in DinG. The scheme on the left illustrates the assay setup. Coloring in the bar graph according to DinG domain coloring in (**b**). *n* = 3 independent replicates; mean ± s.d. Residues producing interference defects upon substitution are highlighted in bold in panel (**c**). Source data are provided as a Source Data file.

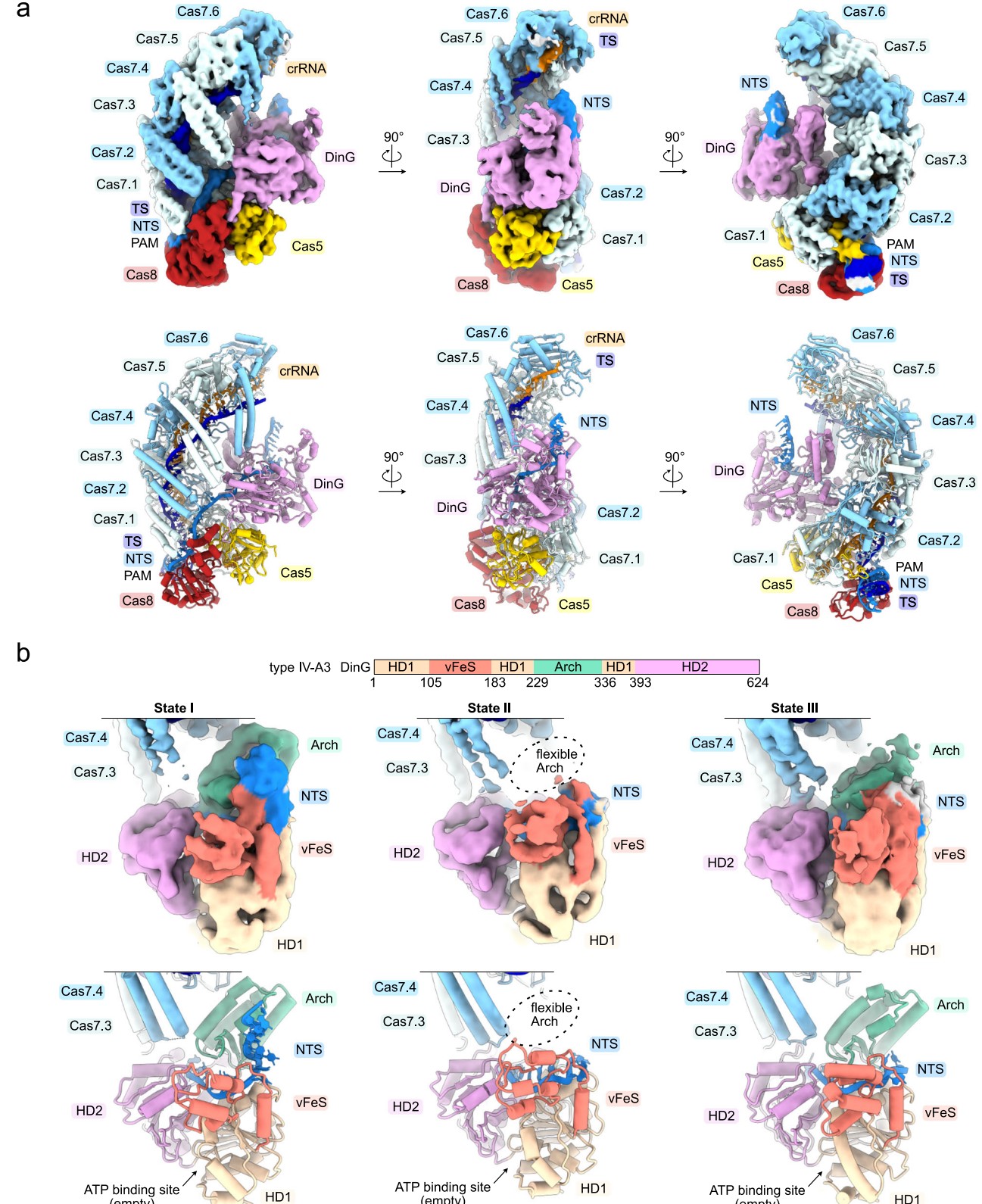

**Fig. 7 | Cryo-EM structures of the type IV-A3 effector bound to DinG. a** Top: Sharpened experimental cryo-EM map (state I) of the type IV-A3 effector in complex with DinG in three orientations. Unfiltered and EMReady maps are shown in Supplementary Fig. 19. Below: Structure model of the type IV-A3 effector in complex with DinG. **b** Top: domain organization scheme of DinG. Below: DinG-centered view of sharpened experimental cryo-EM maps and models of states I, II and III. DinG domains are color coded according to the scheme above.

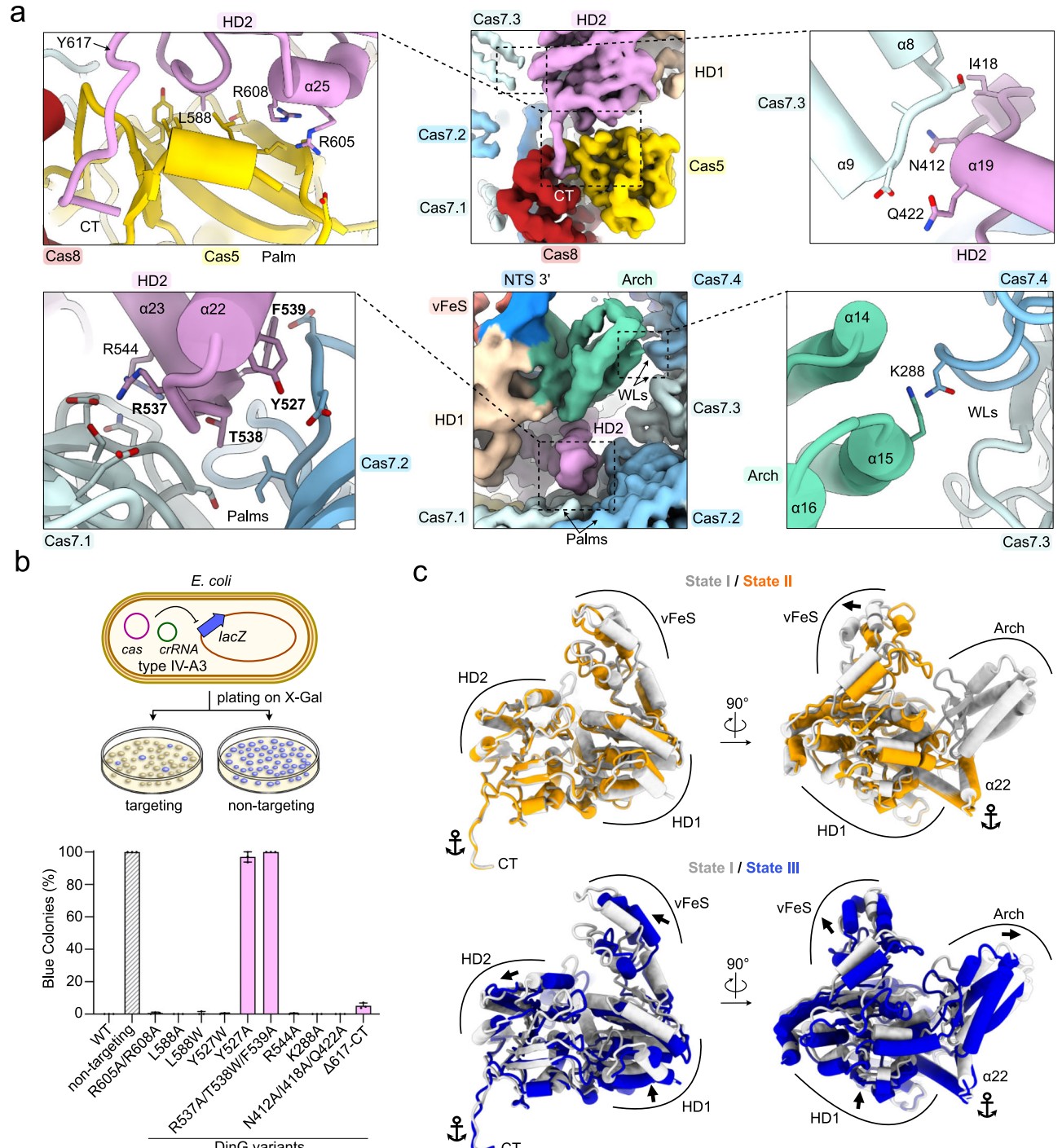

**Fig. 8 | Type IV-A3 DinG interface. a** Close-up views onto the DinG interfaces. The sharpened experimental cryo-EM map is shown in the center. Dashed lines highlight the position of individual interfaces, shown in the left and right insets. DinG residues are shown and labeled. Residues in proximity to DinG are shown and not labeled for clarity. **b** *lacZ*-CRISPRi assay probing amino acid substitutions in DinG. The scheme on top illustrates the assay setup. Coloring in the bar graph according to DinG domain coloring in Fig. 7b. *n* = 3 independent replicates; mean ± s.d. Residues producing strong interference defects upon substitution are highlighted in bold in panel **a**. Source data are provided as a Source Data file. **c** Comparison of DinG states, relative to the complex. The complex is not shown for clarity. Anchor symbols indicate elements that do not rearrange between all states. Arrows indicate rearrangements between individually compared states.

DNA for the type IV-A3 ΔDinG complex, when compared to the type IV-A1 ΔDinG complex (Fig. 4e, f).

We next compared the three type IV-A3 DinG states, based on a structural alignment of the complexes (Fig. 8c). This revealed that the Cas8 and Cas5 contacting C-terminus, and Cas7 palm binding segments of α-helices 21 and 22 of DinG, do not rearrange between the states (Fig. 8c). Thus, the C-terminus and α-helices 21 and 22 appear to

anchor DinG to the complex. Between states I and II, the HD domains maintain their relative and overall conformations, while the vFeS moves towards the center of DinG (Fig. 8c). The arch domain becomes stabilized in state I, as corroborated by the absence and presence of cryo-EM density in states II and I, respectively (Fig. 7b). Comparing states I and III revealed that the overall conformation of HD1, HD2 and vFeS are maintained, though they reposition in concert (Fig. 8c).

Additionally, the arch domain rearranges and becomes stabilized in state I, as evidenced by the less pronounced cryo-EM density in state III (Fig. 7b). Examination of the cryo-EM map also revealed low local resolutions (4–7 Å) for the vFeS domain and NTS exiting the HD2 in all states (Supplementary Fig. 18). This limits the accuracy of structural interpretations and suggests an inherent flexibility within these regions. Domain flexibility may facilitate the dynamic recruitment of DinG to the NTS and regulation of its helicase activity. However, the precise order of events and regulation thereof remain to be determined.

## Discussion

Our study revealed an unexpected degree of structural diversity between type IV-A1 and type IV-A3, despite the presence of an equal set of Cas proteins. Our data support a model for crRNA-guided DNA-surveillance that is analogous to models for type I systems and type IV-A1 from *P. aeruginosa*[20,21] (Fig. 9): DNA binding begins with PAM recognition by Cas8 and Cas5, leading to local unwinding of the dsDNA downstream of the PAM for hybridization of the TS and crRNA along the Cas7 backbone. Parallel to sequence interrogation, Cas8 and Cas7 direct the NTS towards the DinG interface. Upon R-loop formation, NTS display promotes DinG recruitment at the center of the complex, where its helicase domains then load onto the NTS.

The structures of types IV-A1 and IV-A3 in complex with DinG suggest a mechanism where DinG's HD1 and HD2 helicase core first engages the NTS, and then the vFeS domain rearranges to position the NTS for association with the arch domain. An alternative interpretation of our observed conformational states posits that the arch domain dissociates after NTS loading, leaving the exact sequence of these events open for further investigation. Notably, the arch domain of the related human XPD helicase forms contacts with other proteins in the human transcription factor IIH complex[45,46], possibly for helicase regulation[15]. After assuming the ATPase-competent conformation, ATP binding and hydrolysis must stimulate DinG's helicase activity, promoting 5′-3′ DNA-translocation for transcriptional interference. In turn, DinG release may permit recruitment of another DinG to the effector, enabling multiple turnovers (Fig. 9).

In remotely related type I CRISPR systems, DNA-target binding and R-loop formation results in large conformational changes in Cas11 and Cas8, which lead to Cas3 recruitment, DNA-unwinding and degradation[21]. Our structures revealed contacts between DinG and Cas7 in *P. oleovorans* type IV-A1 and additional interfaces between DinG and Cas8, Cas5, as well as Cas7 in *K. pneumoniae* type IV-A3. These interfaces might only become available for DinG recruitment upon R-loop completion; future studies should address the binary state structures and conformational dynamics to understand the

mechanistic details. Noteworthy, the binary- and ternary-state structures of the *P. aeruginosa* type IV-A1 effector revealed a ~10 Å-wide condensation, likely triggered by R-loop formation[20]. Concomitant to binding of the DNA, Cas8, Cas5, and the Cas7 wrist loops (forefingers) and fingers rearrange (Supplementary Fig. 21), possibly signaling for DinG recruitment at the wrist-loop interface.

The structural and mechanistic nuances of CasDinG helicases provide insights into the diversification of type IV CRISPR-Cas systems and a broader understanding of CRISPR-Cas system evolution. Our findings reveal that type IV-A3 DinG lacks the predicted N-terminal DNA-binding domain found in type IV-A1 DinG, which is dispensable for helicase activity[15]. This suggests that the N-terminal domain may not be required for transcriptional interference but may instead carry out alternative functions, such as regulation. Interestingly, the N-terminal domain of type IV-A1 DinG is susceptible to proteolysis, though the biological implications remain unclear[47]. Our study also reports marked differences in types IV-A1 and IV-A3 strategies for DinG recruitment, PAM-identification and DNA binding, despite their close relatedness and similar transcriptional interference activities[12,18]. Given that Acrs are important drivers of CRISPR-Cas system component diversification[48], we speculate that DinG- and PAM-recognition, as well as DNA interaction interfaces are common targets for type IV-A inhibition. Yet, no Acrs have been identified against type IV systems to date.

In some rare type IV-A variants, the role of DinG has further diversified through the acquisition of an HNH nuclease domain[11]. This adaptation enables targeting activities similar to Cas3 in type I systems, highlighting the convergent evolution of helicase-nuclease functions in highly divergent CRISPR-Cas systems (types I and IV) and demonstrating the evolutionary plasticity of CRISPR-Cas systems through protein domain shuffling. Further investigation could determine if the evolution of nucleolytic targeting in these systems serves functions beyond the primary plasmid-plasmid competition role reported in type IV-A systems[6,12]. Understanding these evolutionary adaptations will enrich our knowledge of type IV-A evolution and lay the foundation for engineering advanced CRISPR-based technologies.

In conclusion, the compact nature of type IV-A systems, in contrast to many type I systems, renders them appealing for genome editing, especially when the cargo capacity of genome editing vectors, such as AAV, becomes limiting. Our structural and mechanistic insights should enable the engineering of type IV-A-based genome editors for transcriptional interference. Moreover, our findings can guide the design of next-generation genome editors, functionalized by enzymes and proteins for base editing[49], CRISPRi/a[50], and epigenome editing[51], providing this compact system for future therapeutic and agricultural use.

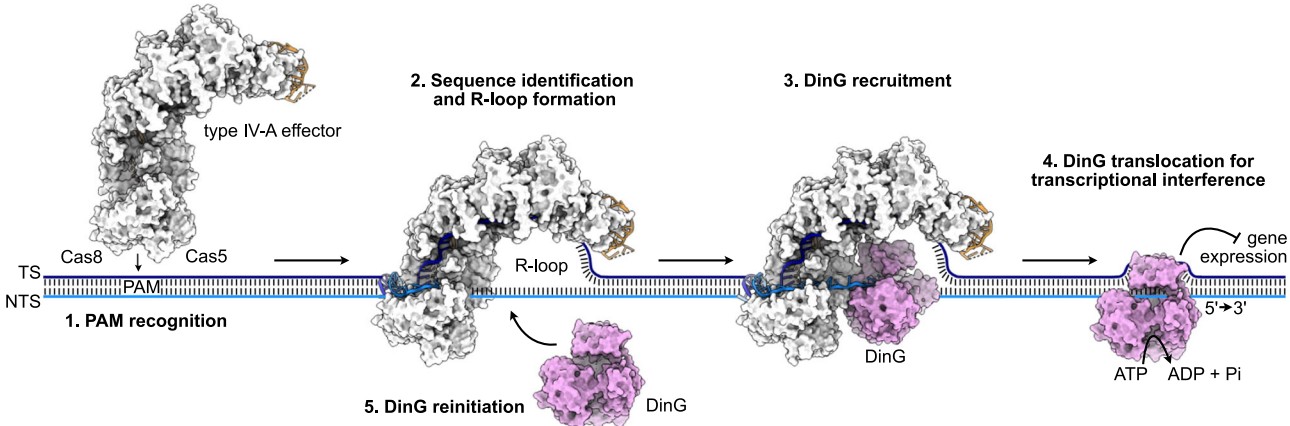

**Fig. 9 | Model of type IV-A–mediated interference.** The type IV-A1 effector surface model is shown exemplarily.

## Methods

### Cloning, expression, and purification of types IV-A1 and IV-A3 complexes

Type IV-A1 *cas* genes, including a C-terminal hexa-histidine-tag encoding fusion of *cas8*, were cloned into the first multiple cloning site of a with EcoRV and HindIII digested pRSFDuet-1 via Gibson cloning. The same method was used to insert *dinG* to the second multiple cloning site of the same vector digested with NdeI and XhoI. The pUC19 vector encoding the type IV-A1 mini-CRISPR array consisting of repeat-spacer-repeat was synthesized by Genscript (Piscataway, NJ, USA). The cloning strategy for type IV-A1 effector complex (*cas7-his6*) and gRNA encoding expression plasmids is described in Benz et al., 2023[12]. Types IV-A1 and IV-A3 *cas*-operon and crRNA encoding plasmids were co-transformed in *E. coli* BL21 Star (DE3) (Novagen) and plated on LB-agar plates containing carbenicillin (50 μg/mL) or ampicillin (100 μg/mL), in addition to kanamycin (50 μg/mL) and incubated at 37 °C. The effector complexes were expressed and purified as previously described for type IV-A3[12]. In brief, to express the type IV-A3 ribonucleoprotein complexes, 15 mL overnight cultures (TB-carb-kan) were used to inoculate 1 L TB medium (TB-carb-kan) and incubated shaking vigorously at 37 °C to an $OD_{600}$ of 0.6–0.8. Gene expression was induced with 0.5 mM IPTG, and cultures were grown for an additional 3 h at 37 °C (type IV-A3). To express type IV-A1 ribonucleoprotein complexes, 500 ml LB medium (LB-amp-kan) was inoculated with a fresh overnight culture to an $OD_{600}$ of 0.1. Cells were grown at 37 °C until reaching an $OD_{600}$ of 0.6–0.8. Gene expression was induced with 1 mM IPTG and cells were grown overnight at 18 °C. Cells were subsequently harvested and resuspended in 20 mL lysis buffer (10 mM HEPES-Na, pH 8.0, 150 mM NaCl, 40 mM Imidazole). Cells were lysed by sonication using a Vibra-Cell ultrasonic processor at 40% amplitude for 5 min with pulses of 3 s at 3 s intervals, before lysate clarification via centrifugation ($47,384 \times g$, 20 min at 4 °C). Supernatants were applied to 5 mL HisTrap FF columns (Cytiva), pre-equilibrated in lysis buffer at 4 °C. After a wash step with 15 column volumes lysis buffer, bound complexes were eluted with three volumes elution buffer (10 mM HEPES-Na, pH 8.0, 150 mM NaCl, 500 mM Imidazole). Concentrated Type IV-A1 complexes were further purified by size exclusion using a Superose 6 Increase 10/300 GL column (Cytiva), equilibrated in size exclusion buffer (10 mM HEPES-Na, pH 7.5, 150 mM NaCl) at 4 °C, Type IV-A3 complexes were further purified by size exclusion using a Superdex 200 Increase 10/300 GL column (Cytiva), equilibrated in the same size exclusion buffer at 4 °C. Complexes were concentrated to approximately 0.5 mL and concentrations were estimated based on the absorbance at 280 nm using a NanoDrop Eight spectrophotometer (Thermo) and extinction coefficients derived from Cas protein stoichiometries of 1:1:5:1 (Cas5:Cas8:Cas7:Cas6) for type IV-A1 and 1:1:6:1 (Cas5:Cas8:Cas7:Cas6) for type IV-A3.

### Analytical size-exclusion chromatography

In brief, mutant and WT *dinG-his6* were either cloned into pRSFDuet-1 (type IV-A1), or pET20b (type IV-A3) vectors and expressed in *E. coli* BL21-AI or *E. coli* BL21 Star (DE3) (Novagen), respectively. Cells were cultured in LB-Amp (100 μg/mL ampicillin) or in LB-Kan (50 μg/mL kanamycin). Gene expression was induced at $OD_{600}$ of 0.6–0.8 with 0.5–1 mM IPTG and cells were grown overnight at 18 °C. Proteins were purified by Ni-NTA affinity chromatography and preparative SEC, according to the strategy described above, using the following buffers for type IV-A1 *dinG-his6*: Lysis buffer (50 mM Tris-HCl pH 8.0, 20 mM Imidazole, 300 mM NaCl), elution buffer (50 mM Tris-HCl pH 8.0, 500 mM Imidazole, 300 mM NaCl) and SEC buffer (50 mM Tris-HCl pH 8.0, 300 mM NaCl) and the following for type IV-A3 *dinG-his6*: Lysis buffer (10 mM HEPES pH 8.0, 40 mM Imidazole, 500 mM NaCl), elution buffer (10 mM HEPES pH 8.0, 500 mM Imidazole, 500 mM NaCl) and SEC buffer (10 mM HEPES pH 8.0, 500 mM NaCl). For the analytical SEC, 10 nmol protein was run on a Superose 6 Increase 10/300 GL column (Cytiva) in SEC buffer (50 mM Tris-HCl pH 8.0, 300 mM NaCl or 10 mM HEPES pH 8.0, 500 mM NaCl).

### Complex reconstitution for cryo-EM

DNA oligonucleotides (Supplementary Table 2) were designed to contain a non-complementary protospacer segment between target and nontarget DNA strands to produce 'bubbled' dsDNA substrates and facilitate rapid R-loop formation during ternary complex reconstitution. Oligonucleotides were synthesized by Metabion. DNA oligonucleotides were combined in a 1:1.5 molar ratio (target strand:nontarget strand) and annealed to a final DNA duplex concentration of 0.5 mM in SEC buffer (10 mM HEPES-Na pH 7.5 RT, 150 mM NaCl) by heating the reaction mixture for 5 min at 95 °C and a subsequent slow cool down (0.1 °C/s) in a thermocycler. Types IV-A1 and IV-A3 ternary complexes were reconstituted by incubation of 10 μM type IV effector complexes and 15 μM target DNA for 10 min at RT in a total volume of 200 μl SEC buffer. Subsequently, Type IV-A1 assembly reactions were injected into a Superose 6 Increase 10/300 GL column (Cytiva) pre-equilibrated in SEC buffer at 4 °C to separate complexes from excess nucleic acids, the same procedure was performed with type IV-A3 assembly reactions using a Superdex 200 Increase 10/300 GL column (Cytiva). Peak fractions were concentrated to 200 μl at 4 °C and concentrations were estimated using the absorbance at 280 nm, as measured on a NanoDrop 8000 Spectrophotometer (Thermo Scientific). Assembled complexes were kept on ice to prevent aggregation.

### Cryo-EM grid preparation and data collection

For the types IV-A1 and IV-A3 effector complexes bound to target-DNA and DinG, cryo-EM grids were prepared by applying 3 μL of sample (0.5 mg/mL) to a glow discharged (20 mA for 7 s using GLOQUBE PLUS (Quorum)) Quantifoil R 1.2/1.3 on 300 copper mesh grid with a 2 nm continuous carbon support layer. For type IV-A3 with target-DNA in absence of DinG, cryo-EM grids were prepared by applying 3 μL of sample (0.5 mg/mL) to a Quantifoil R 1.2/1.3 300 copper mesh grid, glow discharged at 20 mA for 45 s using GLOQUBE PLUS (Quorum). The grids were flash frozen in liquid ethane using Vitrobot Mark IV (Thermo Scientific), at 4 °C, 100% relative humidity with no preincubation and 6 s blot time. Samples were imaged on a 200 kV Glacios Cryo-TEM (Thermo Scientific) equipped with a Falcon III direct electron detector (Thermo Scientific). Movies of the samples were collected at a nominal magnification of ×92,000, at a calibrated pixel size of 1.1 Å, 30 frames and a total dose of 30 $e^-$/Å$^2$, with a defocus range of −2.0 to −1.0 μm.

### Cryo-EM data processing and 3D variability analysis

Motion correction, CTF estimation, and blob particle auto-picking were performed on the fly in CryoSPARC live (v. 4.2)[52]. Subsequent steps of 2D classification, initial model generation and refinement were performed using CryoSPARC (v. 4.2)[52]. Data for type IV-A3 in presence of DinG was processed using CryoSPARC (v. 4.4)[52].

A dataset of 2333 movies was collected for the type IV-A1 sample. The blob auto-picking resulted in 1,022,768 picked particles, which were extracted with a box size of 360 pixels. After several rounds of 2D classification and heterogeneous refinement, a set of good particles of 159,623 resulted in classes where a density corresponding to DinG was present or absent. The particles were then refined with per particle CTF and motion correction. The complex with DinG was additionally masked to improve the resolution of DinG and Cas6, resulting in a map with DinG at an overall resolution 3.2 Å, of isolated DinG at 4.2 Å and Cas6 at 4.8 Å. The map for DinG was sharpened using the sharpening tools job in Cryosparc (v4.4). The type IV-A1 complex without DinG was refined to a final resolution of 2.96 Å. Composite maps were generated by aligning the locally refined maps to the main maps and using the vop maximum command in UCSF ChimeraX (v. 1.8)[53]. The maps used for initial modeling were further processed using EMReady[54].

For the type IV-A3 sample in absence of DinG, a dataset of 1120 movies was collected and pre-processed using the protocol as described above for type IV-A1, resulting in an initial set of 1,082,094 particles. After a round of 2D classification to clean up junk particles, and a multi class of ab-initio reconstruction, the best class was picked, containing 448,275 particles. Heterogeneous classification was used to remove noisy particles from the reconstruction, and the best reconstruction was used for 3D classification with 3 classes. The class showing the most distinct features and most interpretable PAM distal region was chosen for final per particle CTF and local motion correction. The resulting reconstruction of 104,035 particles was then refined, resulting in a final overall resolution of 2.8 Å. The map used for initial modeling was processed using EMReady[54].

A dataset of 4350 movies was collected for the type IV-A3 complex bound to DinG. The blob auto-picking resulted in 4,069,893 picked particles, which were extracted with a box size of 360 pixels. After a round of 2D classification to clean up junk particles, and a multi class of ab-initio reconstruction, a set of 729,551 good particles was selected for further processing. Heterogeneous classification was used to remove particles, containing partial or no DinG. The reconstruction containing DinG (140,065 particles) was used for 3D classification with 3 classes. Three distinct states of DinG were obtained by 3D classification. The final polishing was applied to all 3 states by reference-based motion correction, per particle CTF and local motion correction. The resulting reconstructions of 3 states were then refined, resulting in a final global resolution of 2.9 Å (state I; 45,309 particles), 2.88 Å (state II; 48,197 particles), 2.89 Å (state III; 46,284 particles). The final maps were further processed using EMReady[54].

For 3D variability analysis (3DVA) in CryoSPARC (v. 4.2)[52], the reconstruction of type IV-A1 without DinG (100,000 particles) was used with three modes of motion and a resolution cutoff of 10 Å. The three modes were visualized by 3DVA simple display using 20 frames for each component. For type IV-A3, the initial pool of picked particles was sorted in 3D to obtain a consensus map (~390,000 particles), which was then subjected to 3DVA using the same settings as for type IV-A1.

## Model building, refinement, and figure preparation

Models were built into EMReady processed maps, and double checked and refined against the sharpened experimental maps. For type IV-A3, Initial models of protein subunits were computationally predicted using AlphaFold[55], and manually re-built using Coot (v. 0.9.8)[56] and ISOLDE[57] in ChimeraX (v. 1.4)[58]. The type IV-A1 initial model was generated using ModelAngelo[59] and modeled using Coot (v. 0.9.8) and the ISOLDE suite in ChimeraX (v. 1.8). Models were refined using Phenix (v. 1.19.2)[60] real space refinement against the sharpened composite maps. Secondary structure restraints were generated for the protein, and manual restraints for the base pairing between the visible nucleic acid base pairs were added. For refinement of type IV-A1 without DinG and type IV-A3 with and without DinG, atomic displacement, global minimization and local grid search strategies were used. For type IV-A1 with DinG, in addition to the previous strategies, simulated annealing was also employed. Figures were prepared in UCSF ChimeraX[53], UCSF Chimera[54] and Coot[49].

## Mass photometry

Samples were prepared according to the procedure described for cryo-EM complex reconstitution. crRNP complexes (~100 nM) were analyzed in presence and absence of DNA on a TwoMP mass photometer (Refeyn Ltd., Oxford). Microscope coverslips (1.5 H, 24 × 60 mm, Carl Roth) and CultureWell Reusable Gaskets (CW-50R-1.0, 3 × 1 mm, Grace Biolabs) were cleaned with three alternating rinsing steps of ddH$_2$O and 100% isopropanol, and dried under a stream of compressed air. Silicone gaskets with six cavities were adhered on coverslips and mounted on the stage of the mass photometer using immersion oil (ImmersolTM 519 F, Carl Zeiss, Jena). Prior to each measurement, 18 µl of phosphate-buffered saline (137 mM NaCl, 2.7 mM KCl, 12 mM phosphate, pH 7.4, RT) was pipetted into one cavity, and the instrument was focused. 2 µL of protein samples were added, mixed, and measured for 60 s at 100 frames/s using AcquireMP (Refeyn Ltd., v.2023 R1.1). Measurements were repeated at least three times with similar results. The instrument was calibrated using an in-house made calibration standard of a protein mixture with known sizes (86–430 kDa), and data was fit to a linear regression. All data was analyzed using DiscoverMP (Refeyn Ltd., v.2023 R1.2).

## Phage targeting assay

The functionality of type IV-A3 mutants was evaluated by targeting phage λ-vir in gene *b* as previously described in a phage-spotting assay[12]. In brief, to assess the replication of CRISPR-targeted phage λ-vir on bacterial lawns (GeneHogs) compared to a NT control, the type IV-A3 *cas* operon from the Ptac promoter and crRNAs from the PBad promoter were expressed during overnight growth in LB-Amp-Gent (ampicillin 100 µg/mL, gentamicin 20 µg/mL) in triplicates of *E. coli* GeneHogs. We combined 150 µL of bacterial overnight cultures with 4 mL of molten top agar (0.7% w/v) containing 10 mM MgSO$_4$, L-arabinose (0.3% w/v), and IPTG. This mixture was poured onto LB agar plates with added MgSO$_4$, L-arabinose, and IPTG, and then 5 µL of 10-fold serially diluted phage lysates were spotted onto the lawn. Plates were incubated at 30 °C and plaque forming units (PFU) were counted as a measure of phage replication the following day.

## Efficiency of transformation assay

*E. coli* BL21-AI cells were used to express recombinant type IV-A1 effectors using a pETDuet-1 plasmid encoding all *cas* genes and a pCDFDuet-1 carrying a minimal CRISPR array with crRNA1 of *P. oleovorans*. Mutations in the crRNAs and protospacer sequence were generated via Quikchange site-directed mutagenesis, or by ligating two phosphorylated and hybridized oligonucleotides containing the mutated sequence into a prior digested vector. Cells producing type IV effectors containing one of the crRNAs (no mismatch or variable mutations of bases) were transformed with pACYCDuet-1 either carrying a perfect target with 5′-AAG-3′ PAM or a random spacer sequence. To test protospacer mutations, pACYCDuet-1 carrying desired sequences were transformed. As a control, cells expressing wild type spacer1 were transformed with target and non-target plasmid. Transformation efficiency was calculated with the formula: Transformation efficiency = c.f.u. (sample)/c.f.u. (non-target control), as previously described in Guo et al.[9].

## gfp reporter gene assay in P. oleovorans

*crRNA*-encoding pSR106 plasmids were introduced into *gfp*-encoding *P. oleovorans* DSM1045 via conjugation, as previously described[9]. In brief, pSR106 plasmids were transformed into the DAP-auxotroph helper strain *E. coli* WM3064, plated on LB-agar plates containing spectinomycin (100 µg/mL) and incubated at 37 °C. In the first step of conjugation, overnight cultures of *P. oleovorans* and *E. coli* WM3064 cells were grown at 37 °C in liquid LB media; cultures of *E. coli* WM3064 cells were supplemented with spectinomycin (100 µg/mL) and DAP (0.3 mM). Overnight cultures were collected separately by centrifugation and washed twice with LB media supplemented with 0.3 mM DAP. Cells were resuspended in 100 µl LB-DAP (0.3 mM DAP) and mixed before spot plating onto LB-DAP (0.3 mM DAP) agar. After incubation for 5–7 h at 37 °C, cells were resuspended in 2 mL of LB. Cells were washed twice with LB to remove the residual DAP resulting in the elimination of *E. coli* WM3064. Serial dilutions were plated on agar containing spectinomycin (100 µg/mL) and incubated at 37 °C for 36 h. Conjugants were inoculated in 3 mL LB media supplemented with spectinomycin (100 µg/mL) and were grown shaking overnight at 37 °C. Cells were inoculated to an OD$_{600}$ of 1 in a 96-well plate and

grown at 37 °C in 200 μL of LB media supplemented with 0.1 mM IPTG and 0.5% arabinose for induction of *crRNA-* and *gfp-* expression, respectively. Fluorescence of GFP and $OD_{600}$ were measured over time (48 h) using a plate reader (Tecan Infinite 200 Pro, $\lambda_{ex}$: 485 nm and $\lambda_{em}$: 510 nm). Data were background subtracted and GFP fluorescence was normalized with the corresponding $OD_{600}$ values for each cycle. Mean values were plotted using R-studio (v.2023.03.0 + 386).

## Beta galactosidase repression assay

*E. coli* BL21-AI (Thermo Fisher) were transformed with a plasmid encoding all type IV-A1 *cas* genes and a minimal CRISPR array harboring a spacer sequence targeting *lacZ*, or with two plasmids one encoding all type IV-A3 genes and the other a minimal CRISPR array comprising a spacer for *lacZ*. Fresh transformants were inoculated in 2 mL of LB media supplemented with kanamycin (50 μg/μL) for type IV-A1, or kanamycin (50 μg/μL) and (100 μg/mL ampicillin) for type IV-A3, and grown for 4 h at 37 °C shaking. All cultures were brought to an $OD_{600}$ of 1 and dilutions ranging from $10^{-1}–10^{-5}$ were plated on LB-Kan or LB-Kan/Amp (50 μg/mL kanamycin, 100 μg/mL ampicillin) plates with arabinose (0.2%), IPTG (1 mM) and X-Gal (40 μg/mL), prior to incubation overnight at 37 °C and followed by 3 days at 4 °C for color development. Plates were imaged and analyzed with OpenCFU (v.3.9.0) where the color filter was set to a hue angle of 0–50 to differentiate between blue and white colonies, or counted to determine the percentage of blue colonies.

## Electrophoretic mobility shift assay

Plasmids pRC040-pRC049 were used as templates to amplify linear DNA substrates. Plasmid templates were constructed by mutating pACYCDuet-1 to introduce mutations. The EMSA DNA probe was prepared by amplifying the 365 bp-length DNA-target sequence, encoding the mutated *cat* promoter, using ATTO647N-labeled primers. The PCR product was verified by agarose gel electrophoresis and gel-extracted. The purified DNA was quantified using a NanoDrop Eight spectrophotometer. EMSA experiments were carried out by preparing 12 μL reactions containing 1 nM ATTO647N-labeled DNA substrate and varying concentrations of type IV-A ΔDinG complexes in SEC buffer (10 mM HEPES pH 7.5, 150 mM NaCl). Reactions were assembled on ice and incubated for 20 min at RT. After incubation, 3 μL of 5X Orange G sample loading dye (0.025% Orange G dye (v/v), 50% glycerol, 0.5 M EDTA pH 8) was added. The reactions were then separated on 6% native polyacrylamide gels, which were prepared using a solution of 30% acrylamide/bisacrylamide (29:1), 1X TBE buffer (89 mM Tris, 89 mM boric acid, 2 mM EDTA, pH 8.0), 0.1% ammonium persulfate, and 0.1% TEMED. Electrophoresis was performed at 4 °C using a vertical gel apparatus with 1X TBE running buffer, at a constant voltage of 100 V. Gels were imaged using FLA-5100 (FujiFilm) and Amersham Typhoon (GE Healthcare) imaging systems. The bands corresponding to free DNA and DNA-protein complexes were quantified using ImageJ (version 1.53k). The bound fraction was derived by calculating the ratio of the intensity of the DNA-protein complex band to the total intensity of both the free DNA and DNA-protein complex bands. Curves were fitted using a sigmoidal four-parameter logistic curve model in PRISM (v9.4.1, GraphPad).

## DinG sample preparation for mass spectrometric analysis

Coomassie stained (0.4% Coomassie Brilliant Blue G250, 10% citric acid, 8% ammonium sulfate, 20% methanol) SDS-PAGE gel pieces containing DinG protein were dehydrated by the addition of 100% acetonitrile. Enough volume of 10 mM dithiothreitol (DTT, Sigma) in 50 mM $NH_4HCO_3$ to cover the pieces was added and incubated at 56 °C for 30 min. For destaining, 100% acetonitrile was added and incubated for another 15 min at RT, followed by another dehydration step. Alkylation was performed with 55 mM chloroacetamide (Sigma) in 50 mM $NH_4HCO_3$ for 30 min at RT, followed by dehydration with 100%

acetonitrile. For overnight digestion, 100 ng sequencing-grade trypsin (Promega) was added in 50 mM $NH_4HCO_3$. Peptide extraction was done by sonication for 15 min, followed by centrifugation and supernatant collection. A solution of 50:50 water:acetonitrile, 1% formic acid was used for a second extraction. The supernatants of both extractions were pooled and dried in a vacuum concentrator. Peptides were dissolved in 10 μL of the reconstitution buffer (96:4 water: acetonitrile, 1% formic acid) and analyzed by LC-MS/MS.

## Mass spectrometry of DinG

For LC-MS/MS measurement, an Orbitrap Fusion Lumos instrument (Thermo) coupled to an UltiMate 3000 RSLC nano LC system (Dionex) was used. Peptides were concentrated on a trapping cartridge (μ-Precolumn C18 PepMap 100, 5 μm, 300 μm i.d. × 5 mm, 100 Å) with a constant flow of 0.05% trifluoroacetic acid in water at 30 μL/min for 4 min. Subsequently, peptides were eluted and separated on the analytical column (nanoEase™ M/Z HSS T3 column 75 μm × 250 mm C18, 1.8 μm, 100 Å, Waters) using a gradient composed of Solvent A (3% DMSO, 0.1% formic acid in water) and solvent B (3% DMSO, 0.1% formic acid in acetonitrile) with a constant flow of 0.3 μL/min. The percentage of solvent B was stepwise increased from 2% to 8% in 6 min, to 25% for a further 6 min, to 40% in another 3 min and to 85% in 8.5 min, and back to 2% in 2.5 min. The outlet of the analytical column was coupled directly to the mass spectrometer using the nanoFlex source equipped with a Pico-Tip Emitter 360 μm OD × 20 μm ID; 10 μm tip (CoAnn Technologies). Instrument parameters: spray voltage of 2.4 kV; positive mode; capillary temperature 275 °C; mass range 350–1500 m/z (Full scan) in profile mode in the Orbitrap with resolution of 120,000; Fill time 100 ms with a limitation of 4e5 ions. Data dependent acquisition (DDA) mode, MS/MS scans were acquired in the Iontrap with rapid scan rate, with a fill time of up to 35 ms and a limitation of 1e4 ions (AGC target). A normalized collision energy of 30 was applied (HCD). MS2 data was acquired in centroid mode. Xcalibur (v4.7, Thermo Fisher Scientific) was used for data acquisition.

## Mass spectrometry data processing

The raw mass spectrometry data was processed with MSFragger pipeline (MS Fragger v3.8, Fragpipe v20, IonQuant v1.9.8, Philosopher v5.0.0)[61] and searched against the uniprot-proteome UP000000625 (*E. coli*, strain K12, 4402 entries, October 2022) database including common contaminants, reversed sequences and the sequence of DinG. The data was searched essentially with default settings and the following modifications: Carbamidomethyl (C) as fixed modification, acetylation (Protein N-term) and oxidation (M) as variable modifications. The default mass error tolerance for the full scan MS spectra (20 ppm) and for MS/MS spectra (0.5 Da) was used. A maximum number of 2 missed cleavages was allowed. For protein identification, a minimum of 1 unique peptide with a peptide length of at least seven amino acids and a false discovery rate below 0.01 were required on the peptide and protein level. Match between runs was enabled with default parameters.

## Commercial reagents

Company names and catalog numbers of commercial reagents are provided as: Supplementary Data 1. Commercial reagents.

## Reporting summary

Further information on research design is available in the Nature Portfolio Reporting Summary linked to this article.

## Data availability

Cryo-EM model coordinates have been deposited in the PDB under accession codes 8RC2, 8RC3, 8RFJ, 8S35, 8S36 and 8S37. The final cryo-EM maps, as well as consensus and focused maps, have been deposited in the EMDB under accession codes EMD-19045, EMD-19046, EMD-

19125, EMD-19688, EMD-19689 and EMD-19690, as well as EMD-19120, EMD-19124, EMD-19126, EMD-19127 and EMD-51026, respectively. Previously published model coordinates used in this study are available at the PDB under accession codes 7XG2, 7JHY, 6H66, 7TRA, 7XEX, 7XF0, 7XF1, 7XG3, 7XG4, 7XG1 and 7XG0. Protein mass spectrometry raw data have been deposited in the ProteomeXchange repository under accession code PXD056399. Source data are provided with this paper.

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

## Acknowledgements

We thank the department of Protein - DNA interactions at the Life Sciences Center at Vilnius University (Virginijus Šikšnys) for access to equipment and the Vilnius University cryo-EM facility (Giedrius Sasnauskas) for support. The authors thank Stephen K. Jones Jr. for critical reading and comments on the manuscript. We further thank Ning Jia for sharing the structure coordinates of the *P. aeruginosa* type IV-A1 system prior to release at the PDB, and the EMBL Proteomics Core Facility for MS-analysis of protein samples. P.P. and L.M. received funding from the European Regional Development Fund under grant agreement number 01.2.2-CPVA-V-716-01-0001 (P.P. and L.M.) with the Central Project Management Agency (CPVA), Lithuania. P.P. receives funding from the Research Council of Lithuania (LMTLT) under grant agreement number S-MIP-22-10 (P.P.), and from the European Molecular Biology Conference (EMBC) under EMBO Installation Grant agreement number 5342-2023 (P.P.). R.P.-R. was supported by the Lundbeck Foundation (grant R347-2020-2346, R.P.R.). F.B. was supported by the SNSF (grants P1EZP3_195539 and P500PB_210944, F.B.). This work was further supported by the DFG-SPP2141 (N.K. and L.R.), and LOEWE Research Cluster Diffusible Signals (L.R.) and the Max Planck Society (N.S. and G.K.A.H.).

## Author contributions

P.P. and R.P.-R. conceived the study, further developed with L.M. and L.R. R.C. and N.K. cloned constructs, purified proteins and reconstituted samples for cryo-EM and mass photometry. G.Z. assisted in the purification of complexes under guidance from R.C. A.M., V.R. and L.M. collected and processed cryo-EM data with input from P.P. A.M., A.S. and P.P. built and refined structure models with input from L.M. N.K. and H.B. performed mismatch tolerance assays. R.C. performed EMSA experiments. N.K., N.S. and G.K.A.H. performed mass photometry. All authors designed experiments and analyzed data. P.P. wrote the manuscript and prepared figures with input from R.C., N.K., A.M., F.B., V.R., S.C.-W., L.R., R.P.-R. and L.M. The manuscript was reviewed and approved by all co-authors.

## Competing interests

The authors declare no competing interests.
