## [Transparent Peer Review file · Nature Communications]

Structural variation of types IV-A1- and IV-A3-mediated CRISPR interference

Corresponding Author: Professor Patrick Pausch

Version 0:

Reviewer comments:

Reviewer #1

(Remarks to the Author)

In this manuscript, Dr. Pausch & colleagues have explored the interference mechanisms of the less studied type IV-A CRISPR-Cas effectors with a focus on the types IV-A1 & A3, through biochemical and structural (Cryo-EM) techniques. Their study provides new insight into the molecular basis of type IV-A mediated DNA-interference, and a structural platform for engineering type IV-A-based genome editing tools. Though Jia's and my labs already published a related study about the type IV-A system (Cui et al., Mol Cell, 2023), the manuscript from Dr. Pausch & colleagues advance our mechanistic understanding of the type IV-A CRISPR-Csf structure and function. The authors gave fair comparisons between the two independent works (this manuscript & Cui's), and further presented new findings as summarized in their abstract section. These works and others should enable the type IV-A CRISPR-Csf complexes to be harnessed as genome-engineering tools for biotechnological applications.

Special comments: Some of the main figures such as Fig-7 are too many details. Some revisions may help readers focus on key components.

Reviewer #2

(Remarks to the Author)

This paper by Čepaitė et al., provides a detailed comparison of the structure and interaction dynamics of DinG helicases within type IV-A1 and IV-A3 CRISPR-Cas systems. The authors examine how DinG is recruited and interacts with various components of these complexes, highlighting differences in their recruitment interfaces and structural conformations. The manuscript offers a comprehensive analysis of the structural differences between DinG in type IV-A1 and IV-A3 systems. There is a previous publication where the structure of Type IV-A system from *P. aeruginosa* has been dissected (Cui et al., Mol Cell 2023). In that paper Cui et al., have provided a variety of functional details contributing to the mechanistic understanding of the type IV-A dsDNA targeting during the type IV-A CRISPR-Cas immunity.

Many of the structural details are very similar to those described in this manuscript. The main novel point of this work are the interactions of the DinG helicase with the cascade-like complexes, especially how DinG interacts with different components of the complexes, such as Cas5, Cas7 and Cas8's CTD. The observation of inherent flexibility in certain regions, such as the FeS domain and NTS, and their potential role in the dynamic recruitment of DinG, is a novel contribution. It suggests that the structural adaptability of these components may facilitate their functional roles within the CRISPR-Cas system. This is the main contribution of this work.

The paper describes structural differences and interactions; however, the functional implications of these observations are not clear. How do these structural variations influence the activity and specificity of DinG in the context of CRISPR-Cas systems? Experimental evidence linking these features to functional outcomes would strengthen the paper. For instance, how do these variations in DinG recruitment and interaction affect the efficiency, accuracy, or regulation of CRISPR-Cas-mediated immunity? While the paper suggests that flexibility facilitates DinG's dynamic recruitment, detailing the mechanistic basis of this recruitment, possibly through kinetic studies, could offer deeper insights into how these processes are regulated in vivo.

Major concerns

1.- I do not understand how the helicase DinG contributes to transcription interference and this paper does not provide

further evidence to clarify this point. However, this is the mechanism supported by the authors in Fig. 7. The primary functions of DinG are associated with DNA repair and maintaining genomic integrity, its activity can also impact transcription in scenarios where transcription machinery encounters DNA damage or structures that are difficult to transcribe. Thus, to my understanding the helicase should facilitate transcription. In fact, while not explicitly described for DinG, helicases in general can have roles in preventing or resolving R-loops, which are prone to forming behind active transcription sites. By unwinding these structures, helicases can prevent the stalling of transcription machinery and thereby facilitate transcription. However, it is also true that this activity can also disrupt ongoing transcription processes, especially if the R-loop or a similar structure was stabilizing the transcription complex. I guess this is not the case in type IV-A1 and A3. This point is not clarified by the authors in the model in Fig7. The absence of functional assays or data linking structural variations to biological outcomes leaves a gap in the narrative. The authors must explain, and support experimentally, why they favour this model instead of one where DinG facilitates the presence of ssDNA for other possible processes leading to interference as discussed in Cui et al Mol. Cell 2023.

2.- The supposed effect in transcription is supported by experimental evidence based on reporter or colony assays. These methods for studying transcriptional repression do not offer a direct observation of the targeted mechanism. In addition, given that DinG is involved in key cellular processes such as replication, DNA repair and genome maintenance it is also possible that the observed effect is not due to transcription impairment of the targeted gene but to other of the key processes in which DinG is involved. There is no direct observation of the molecular mechanism to which the authors have attributed the phenotype, see Fig.2E, 4 B-E etc, and others along the manuscript. The authors should experimentally detect levels of mRNA during targeting, i.e. using real-time PCR, which can be used to quantify mRNA levels of the target genes when the type IV complex is overexpressed.

3.- Another example where the authors used indirect ways of testing a molecular mechanism is observed in figure 4. The authors use an in vivo assay in E.coli to test the mismatches. This is an indirect method to test lack of coupling with the evident lack of control over the experimental conditions as the readout is a phenotype not the molecular event that is supposed to be tested. The readout of the assay when using in vitro methods offers more control over experimental conditions and can provide more precise information about the molecular mechanisms affected by mismatches. Therefore, this assay must be performed biochemically to confirm the effect of the mismatches.

4.- The mention of low local resolutions (4-7 Å) for certain domains like the FeS domain and NTS highlights a significant limitation in structural studies. At this resolution, detailed atomic interactions and precise conformations might not be fully resolved, which could limit the accuracy of structural interpretations and functional inferences drawn from these areas. This limitation is crucial because it might affect the reliability of understanding the dynamic nature of DinG recruitment and its interaction with the CRISPR-Cas system components.

5.- In figure 1, 2, 6A and few more through the manuscript, the cryo-EM maps shown are the AI modified maps. The EMready improved maps are shown in many of the figures instead of the sharpened experimental maps. They should be replaced by the experimental maps in order to provide the referee's and the readers with the experimental information. While the AI modified maps are used to improve and facilitate model building, it's not considered good practice to show in the figures the AI improved map. The use of AI to improve cryo-EM maps is aimed to enhance the interpretability of the density obtained from cryo-EM to facilitate model building. However, this does not mean that the modified map is a fully "trustable" version of the experimental map. Therefore, AI-improved maps must be rigorously validated and verified experimentally. I do not see how this is validated in this manuscript. These maps can be used as a visual guide and the last model must be refined against the experimental map. I think this is not clear in this work. This is a major problem as conclusions derived from these maps may not be experimentally supported. The original, unprocessed AI cryo-EM map must be made available in the figure for the readers to allow independent verification and to assess whether the built model is supported by the data. Following this point there are no maps shown in Fig2C, 2D in the PAM region or 6C in the interaction of DinG-Cas7 interface.

6.- While the discussion on the inherent flexibility within the FeS domain and NTS is intriguing, the implications of this flexibility are not fully explored. Flexibility often plays a critical role in protein function, especially in processes like DNA binding and unwinding. However, without deeper analysis or experimental evidence demonstrating how this flexibility contributes to DinG's functionality or recruitment dynamics, this point remains speculative and underdeveloped.

7.- The paper could strengthen its impact by contextualizing the findings within the broader diversity of CRISPR-Cas systems. DinG helicases' structural and functional nuances might offer insights into the evolutionary pressures shaping these systems. A discussion that places these findings within the wider landscape of CRISPR-Cas diversity and adaptation could offer a more comprehensive view of their significance.

Reviewer #3

(Remarks to the Author)

Summary:

CRISPR-Cas systems are adaptive immune systems found in bacteria and archaea, where they protect the host cell from infection by parasitic mobile genetic elements, such as bacteriophages. Protection is mediated by protein effectors (Cas proteins) that are guided to the genetic material of the invader by short RNA guides (crRNA). Upon base-pairing between the crRNA and the target sequence, most CRISPR-Cas systems activate nucleolytic activities to degrade the genetic material of the invader.

Type IV systems, however, do not possess nucleolytic activity, relying instead on transcriptional repression to achieve defense. Repression is mediated by two components: 1) a multi-subunit effector complex that carries the crRNA, and 2) a helicase called DinG. During immunity, it is thought that the type IV effector complex localizes to the cognate DNA target and recruits DinG, which in turn mediates transcriptional interference. Previous structural work on the type IV-A1 system has shown that DinG associates with the effector complex; however, structural work on other effector complexes beyond type IV-A1 is lacking.

In this work, Cepaite et. al. solved the cryo-EM structures of both type IV-A1 and type IV-A3 effector complexes, in the presence and absence of their respective DinG helicases. Broadly, they show that the type IV-A complexes display similarities with the CASCADE complex of type I system; provide structural understanding of how these complexes recognize the protospacer adjacent motif (PAM); and determine how the complex tolerates mismatches between crRNA and the protospacer target. Noteworthy insights include the molecular details concerning PAM recognition by Cas8 and Cas5, crRNA-TS binding through the Cas7 backbone, and non-template strand displacement by Cas8 and Cas7 that allows for DinG recruitment. Uniquely, structural differences between the ways of how type IV-A1 and type IV-A3 associate with DinG were also characterized.

Major comments:

1. Overall, the structural biology work in this manuscript is very convincing. However, a portion of the results appear to have already been observed in a previous type IV-A1 structure of *P. aeruginosa* (Cui et al. *Molecular Cell*, 2023, which the authors cite). The key novelty is the comparison of the type IV-A1 and type IV-A3 structures, and of how they associate with their respective DinGs. Here the paper would benefit from some biochemical and/or molecular biology approaches that validate and complement their structural predictions.

A. The authors hypothesize that the type IV-A1 and type IV-A3 might possess different DNA binding kinetics, given their distinct PAM recognition mechanisms. An EMSA or fluorescent polarization experiment can test this.

B. Unlike type IV-A1, type IV-A3 DinG contacts the palm domains of Cas5, Cas7.1 and Cas7.2, in addition to interacting with the Cas7 WL motifs. Can these interactions be validated by non-structural means as well? For example, by mutating some of these interfaces, would type IV-A3 immunity against lambda-vir get reduced?

C. Does the translocation of DinG differ between A1 and A3?

2. The authors should address why some experiments were only performed with one version of type IV-A and not the other. For example, mismatch tolerance was only done with the A1 system, while anti-phage defense was only performed with the A3 system. Consistent application of the assays would give a better comparison between the two systems.

3. The title of the manuscript reads a bit too general, considering that there is already previous structural work on type IV-A. A better title would focus on the differences between the A1 and A3 system.

Minor comments:

1. Lines 225-226: "This demonstrates that type IV-A3 functions robustly for PAM recognition." It is not clear what that statement means. Does it mean that the complex can recognize the PAM by redundant interactions?

2. Line 294-295: "...only mismatches in position 1, 4 and the combination of 2, 3 and 4 resulted in increased EOTs (Figure 4C)" This sentence is referring to the right panel of figure 4C; however, it is not representative of the figure being referenced. Mismatches in position 1, 4, and 9 results in increased EOTs; additionally, the combination 2, 3 and 4 is not shown in the graph.

3. Line 300-304 would benefit from including the reference to the figure they are referring to, specifically citing Figure 4C.

4. Line 303-305 "Mismatches in position 21-24 might be tolerated as they allow for completion of the R-loop structure, licensing DinG recruitment and thus interference." Is interesting to note that mismatches in 21-24 and 29-32 are tolerated. The authors argue that 21-24 is tolerated because it still allows for R-loop formation, while 29-32 do not bind to the other strand (shown by the structure). However, do the authors have insights on why 25-28 are not tolerated,? The structure found that these nucleotides do bind to the complementary strand, however, if 21-24 is tolerated because it still allows for R-loop formation, it would seem that 25-28 also would allow for R-loop formation, and hence tolerated.

5. Line 364: "Figure 5: Cas8 and Cas5 guide the NTS." The figure title appears misleading, since the text argues that it is Cas8 and Cas7 that guide the NTS.

6. Figure 2E: Using the shield symbol to denote full CRISPR protection is misleading, as there is clear evidence of turbid plaque formation in some of the mutants (e.g. for amino acid substitutions K79A, Y51F, T114A, K119A). A better way to represent turbid plaque data (that lack obvious countable PFUs) is to set the limit of detection to the phage concentration where there are no visible plaques.

Version 1:

Reviewer comments:

Reviewer #2

(Remarks to the Author)

I appreciate the effort of the authors to address my queries, I think they have made a good job. While some aspects of the mechanisms are still unclear I think the new version very has improves substantially.

Therefore I have no further comments and recommend the paper for publication

Reviewer #3

(Remarks to the Author)

The authors address all of my comments in their revisions, and in my opinion, the manuscript is ready to be published.

Reviewer #1 (Remarks to the Author):

In this manuscript, Dr. Pausch & colleagues have explored the interference mechanisms of the less studied type IV-A CRISPR-Cas effectors with a focus on the types IV-A1 & A3, through biochemical and structural (Cryo-EM) techniques. Their study provides new insight into the molecular basis of type IV-A mediated DNA-interference, and a structural platform for engineering type IV-A-based genome editing tools. Though Jia's and my labs already published a related study about the type IV-A system (Cui et al., Mol Cell, 2023), the manuscript from Dr. Pausch & colleagues advance our mechanistic understanding of the type IV-A CRISPR-Csf structure and function. The authors gave fair comparisons between the two independent works (this manuscript & Cui's), and further presented new findings as summarized in their abstract section. These works and others should enable the type IV-A CRISPR-Csf complexes to be harnessed as genome-engineering tools for biotechnological applications.

Special comments: Some of the main figures such as Fig-7 are too many details. Some revisions may help readers focus on key components.

Response: Thank you for your constructive feedback. We have revised the figures to enhance their clarity and comprehensibility.

Reviewer #2 (Remarks to the Author):

This paper by Čepaitė et al., provides a detailed comparison of the structure and interaction dynamics of DinG helicases within type IV-A1 and IV-A3 CRISPR-Cas systems. The authors examine how DinG is recruited and interacts with various components of these complexes, highlighting differences in their recruitment interfaces and structural conformations. The manuscript offers a comprehensive analysis of the structural differences between DinG in type IV-A1 and IV-A3 systems. There is a previous publication where the structure of Type IV-A system from *P. aeruginosa* has been dissected (Cui et al., Mol Cell 2023). In that paper Cui et al., have provided a variety of functional details contributing to the mechanistic understanding of the type IV-A dsDNA targeting during the type IV-A CRISPR-Cas immunity.

Many of the structural details are very similar to those described in this manuscript. The main novel point of this work are the interactions of the DinG helicase with the cascade-like complexes, especially how DinG interacts with different components of the complexes, such as Cas5, Cas7 and Cas8's CTD. The observation of inherent flexibility in certain regions, such as the FeS domain and NTS, and their potential role in the dynamic recruitment of DinG, is a novel contribution. It suggests that the structural adaptability of these components may facilitate their functional roles within the CRISPR-Cas system. This is the main contribution of this work.

The paper describes structural differences and interactions; however, the functional implications of these observations are not clear. How do these structural variations influence the activity and specificity of DinG in the context of CRISPR-Cas systems? Experimental evidence linking these features to functional outcomes would strengthen the paper. For instance, how do these variations in DinG recruitment and interaction affect the efficiency, accuracy, or regulation of CRISPR-Cas-mediated immunity? While the paper suggests that flexibility facilitates DinG's dynamic recruitment, detailing the mechanistic basis of this recruitment, possibly through kinetic studies, could offer deeper insights into how these processes are regulated *in vivo*.

Thank you for your valuable feedback and suggestions. We appreciate the reviewer's recognition of the novel aspects of our study, particularly the interactions of DinG helicases with Cascade-like complexes and their structural flexibility. We have previously demonstrated that both systems function in seemingly identical ways by repressing targeted genes (Guo et al., 2022, Nat. Mic. <https://doi.org/10.1038/s41564-022-01229-2>; Benz et al., 2024, Cell Host & Microbe, <https://doi.org/10.1016/j.chom.2024.04.016>; Sanchez-Londono et al., 2024, bioRxiv, <https://doi.org/10.1101/2024.06.12.598411>). We agree that it is conceivable that the observed differences in the interaction between DinG and the CRISPR RNPs may lead to different recruitment kinetics, and potentially functional differences. We now address the points raised below by including *in vitro* EMSA and *in vivo lacZ*-CRISPRi assays that link structural variations of the complexes to functional outcomes. However, we believe that a detailed mechanistic study on the kinetics and dynamics of DinG, linking them to functional outcomes, is beyond this 9 figure manuscript and warrants a follow-up study.

Major concerns

1.- I do not understand how the helicase DinG contributes to transcription interference and this paper does not provide further evidence to clarify this point. However, this is the mechanism supported by the authors in Fig. 7. The primary functions of DinG are associated with DNA repair and maintaining genomic integrity, its activity can also impact transcription in scenarios where transcription machinery encounters DNA damage or structures that are difficult to transcribe. Thus, to my understanding the helicase should facilitate transcription. In fact, while not explicitly described for DinG, helicases in general can have roles in preventing or resolving R-loops, which are prone to forming behind active transcription sites. By unwinding these structures, helicases can prevent the stalling of transcription machinery and thereby facilitate transcription. However, it is also true that this activity can also disrupt ongoing transcription processes, especially if the R-loop or a similar structure was stabilizing the transcription complex. I guess this is not the case in type IV-A1 and A3. This point is not clarified by the authors in the model in Fig7. The absence of functional assays or data linking structural variations to biological outcomes leaves a gap in the narrative. The authors must explain, and support experimentally, why they favour this model instead of one where DinG facilitates the presence of ssDNA for other possible processes leading to interference as discussed in Cui et al Mol. Cell 2023.

Previous studies have demonstrated that bacterial DinG helicases, which are related to eukaryotic XPD helicases, exhibit variable domain organizations and divergent activities. For example, *E. coli* DinG is an FeS-cluster-containing helicase that translocates along DNA (<https://doi.org/10.7554/eLife.42400>), whereas *S. aureus* DinG is an FeS-cluster-containing helicase fused to a 3'-5' exonuclease, which is proposed to degrade, rather than displace, RNA and DNA strands (<https://doi.org/10.1042/BJ20111903>). The type IV-A CRISPR-associated (Cas) DinG diverges from these chromosomally encoded helicases in both sequence and domain architecture (e.g., it lacks an FeS-cluster and nuclease, <https://doi.org/10.1093/nar/gkad546>), complicating interpretations of CasDinG's mechanism based on those of other DinG enzymes. We have now expanded the introduction to address this important issue.

Concerning the mechanism of CasDinG-mediated interference, we performed RNA-seq on chromosomally encoded *mcherry*, which was targeted by type IV-A3 (Benz *et al.*, 2024, Cell Host & Microbe, <https://doi.org/10.1016/j.chom.2024.04.016>). The experiment showed that *mcherry* transcription is suppressed when targeted by type IV-A3, aligning with our *in vitro* findings on the transcriptional repression of *degfp* in a cell-free transcription-translation assay (Benz *et al.*, 2024, Cell Host & Microbe, <https://doi.org/10.1016/j.chom.2024.04.016>) and the functional findings on gene repression reported in Guo *et al.*, 2022 (Nat. Mic., <https://doi.org/10.1038/s41564-022-01229-2>). Likewise, we could show long-range down-regulation of the chromosomally encoded histidine operon in *E. coli* using RNAseq, when targeted by a heterologously expressed type IV-A1 system (Sanchez-Londono *et al.*, 2024, bioRxiv, <https://doi.org/10.1101/2024.06.12.598411>). In support of a nuclease-independent transcriptional interference mechanism, targeting of chromosomally encoded genes is not toxic to the host and does not result in genomic deletions (Benz *et al.*, 2024, Cell Host & Microbe and Guo *et al.*, 2022 Nat. Mic.). We have now added a sentence to the introduction to clarify the mechanism and function of type IV-A.

We agree with the reviewer that current molecular and mechanistic insights into the precise transcriptional repression mechanism are lacking. Therefore, we prefer not to propose a specific transcriptional interference mechanism in Fig. 9.

Regarding the model proposed by Cui *et al.* in Mol. Cell, 2023, which suggests that DinG-mediated formation of ssDNA leads to nuclease-dependent degradation of targeted plasmids. We believe that the lack of experimental evidence makes this hypothesis unlikely. Nuclease-mediated DNA degradation typically results in detectable deletions within the targeted regions. However, our previous DNA sequencing of type IV-A-repressed genes has shown that type IV-A-mediated interference does not lead to deletions, supporting a nuclease-independent interference mechanism (Guo *et al.*, 2022, Nat. Mic. <https://doi.org/10.1038/s41564-022-01229-2>; Benz *et al.*, 2023, Cell Host & Microbe, <https://doi.org/10.1016/j.chom.2024.04.016>). We now address the discrepancies between both proposed interference mechanisms in the introduction of our revised manuscript.

2.- The supposed effect in transcription is supported by experimental evidence based on reporter or colony assays. These methods for studying transcriptional repression do not offer a direct observation of the targeted mechanism. In addition, given that DinG is involved in key cellular processes such as replication, DNA repair and genome

maintenance it is also possible that the observed effect is not due to transcription impairment of the targeted gene but to other of the key processes in which DinG is involved. There is no direct observation of the molecular mechanism to which the authors have attributed the phenotype, see Fig.2E, 4 B-E etc, and others along the manuscript. The authors should experimentally detect levels of mRNA during targeting, i.e. using real-time PCR, which can be used to quantify mRNA levels of the target genes when the type IV complex is overexpressed.

We would like to point out that there is no evidence that CasDinG may be connected to the same cellular processes as chromosomally encoded non-CRISPR DinG. The CasDinG variant is only present in a specific subset of bacteria and is always associated with CRISPR-Cas systems which makes it highly unlikely to be a factor for key cellular processes. To clarify this, we have added a sentence to the introduction that states “*CasDinG proteins are distinct from chromosomally encoded non-Cas DinG proteins, suggesting a divergent function*”, supported by the findings reported by Taylor *et al.*, 2021, *Frontiers in Microbiology*, <https://doi.org/10.3389/fmicb.2021.671522>. Additionally, we have revised the manuscript to more clearly reference our previous studies, which demonstrate the repression of targeted genes through a nuclease-independent mechanism (Guo *et al.*, 2022, *Nat. Mic.* <https://doi.org/10.1038/s41564-022-01229-2>; Benz *et al.*, 2024, *Cell Host & Microbe*, <https://doi.org/10.1016/j.chom.2024.04.016>; Sanchez-Londono *et al.*, 2024, *bioRxiv*, <https://doi.org/10.1101/2024.06.12.598411>). Notably, in these works, transcriptional repression is also assessed directly by quantifying mRNA levels of target genes via RNA-seq analysis.

3.- Another example where the authors used indirect ways of testing a molecular mechanism is observed in figure 4. The authors use an *in vivo* assay in *E.coli* to test the mismatches. This is an indirect method to test lack of coupling with the evident lack of control over the experimental conditions as the readout is a phenotype not the molecular event that is supposed to be tested. The readout of the assay when using *in vitro* methods offers more control over experimental conditions and can provide more precise information about the molecular mechanisms affected by mismatches. Therefore, this assay must be performed biochemically to confirm the effect of the mismatches.

In our opinion, testing mismatches in a biological setup provides a more informative approach to assessing mismatch effects with respect to their *in vivo* outcomes, which are relevant to the function and potential application of type IV-A as a genome editing tool. However, we agree with the reviewer that some of our discussion on the molecular mechanisms explaining these *in vivo* observations would benefit from direct molecular evidence. We have now included EMSAs that address PAM-distal mismatches and their impact on CRISPR-RNP-DNA interactions.

In a nutshell, the assay revealed that mismatches in the middle of the spacer are less tolerated than mismatches in the PAM-distal part of the crRNA-TS hybrid for type IV-A1 (Figure 4E). We now additionally present *in vitro* mismatch data on type IV-A3 that shows a more stringent mismatch recognition, as compared to type IV-A1 (Figure 4F).

4.- The mention of low local resolutions (4-7 Å) for certain domains like the FeS domain and NTS highlights a significant limitation in structural studies. At this resolution, detailed atomic interactions and precise conformations might not be fully resolved, which could limit the accuracy of structural interpretations and functional inferences drawn from these areas. This limitation is crucial because it might affect the reliability of understanding the dynamic nature of DinG recruitment and its interaction with the CRISPR-Cas system components.

We agree with the reviewer on this point. For this reason, we provided local resolution maps and a complete comparison of all EM maps in the manuscript's supplement. Additionally, we have revised the text to clearly state that the low local resolution limits the accuracy of structural interpretations and suggests flexibility.

5.- In figure 1, 2, 6A and few more through the manuscript, the cryo-EM maps shown are the AI modified maps. The EMready improved maps are shown in many of the figures instead of the sharpened experimental maps. They should be replaced by the experimental maps in order to provide the referee's and the readers with the experimental information. While the AI modified maps are used to improve and facilitate model building, it's not considered good practice to show in the figures the AI improved map. The use of AI to improve cryo-EM maps is aimed to enhance the interpretability of the density obtained from cryo-EM to facilitate model building. However, this does not mean that the modified map is a fully “trustable” version of the experimental map. Therefore, AI-improved maps must be rigorously

validated and verified experimentally. I do not see how this is validated in this manuscript. These maps can be used as a visual guide and the last model must be refined against the experimental map. I think this is not clear in this work. This is a major problem as conclusions derived from these maps may not be experimentally supported. The original, unprocessed AI cryo-EM map must be made available in the figure for the readers to allow independent verification and to assess whether the built model is supported by the data. Following this point there are no maps shown in Fig2C, 2D in the PAM region or 6C in the interaction of DinG-Cas7 interface.

We have shown all experimental maps, including resolution maps, as well as the AI-modified map in the supplement of the manuscript. We now also show the experimental maps in the main text figures. We have additionally reprocessed our cryo-EM data for type IV-A1 in complex with DinG and revised all models (see updated Supplementary Table 1 and 2).

6.- While the discussion on the inherent flexibility within the FeS domain and NTS is intriguing, the implications of this flexibility are not fully explored. Flexibility often plays a critical role in protein function, especially in processes like DNA binding and unwinding. However, without deeper analysis or experimental evidence demonstrating how this flexibility contributes to DinG's functionality or recruitment dynamics, this point remains speculative and underdeveloped.

We agree with this statement. For this reason, we prefer to only describe our observations and avoid speculating on the implications of the observed flexibility.

7.- The paper could strengthen its impact by contextualizing the findings within the broader diversity of CRISPR-Cas systems. DinG helicases' structural and functional nuances might offer insights into the evolutionary pressures shaping these systems. A discussion that places these findings within the wider landscape of CRISPR-Cas diversity and adaptation could offer a more comprehensive view of their significance.

Thank you for this suggestion. We have added a paragraph discussing the implications of our findings in relation to the broader landscape of CRISPR-Cas diversity.

Reviewer #3 (Remarks to the Author):

Summary:

CRISPR-Cas systems are adaptive immune systems found in bacteria and archaea, where they protect the host cell from infection by parasitic mobile genetic elements, such as bacteriophages. Protection is mediated by protein effectors (Cas proteins) that are guided to the genetic material of the invader by short RNA guides (crRNA). Upon base-pairing between the crRNA and the target sequence, most CRISPR-Cas systems activate nucleolytic activities to degrade the genetic material of the invader.

Type IV systems, however, do not possess nucleolytic activity, relying instead on transcriptional repression to achieve defense. Repression is mediated by two components: 1) a multi-subunit effector complex that carries the crRNA, and 2) a helicase called DinG. During immunity, it is thought that the type IV effector complex localizes to the cognate DNA target and recruits DinG, which in turn mediates transcriptional interference. Previous structural work on the type IV-A1 system has shown that DinG associates with the effector complex; however, structural work on other effector complexes beyond type IV-A1 is lacking.

In this work, Cepaite et. al. solved the cryo-EM structures of both type IV-A1 and type IV-A3 effector complexes, in the presence and absence of their respective DinG helicases. Broadly, they show that the type IV-A complexes display similarities with the CASCADE complex of type I system; provide structural understanding of how these complexes recognize the protospacer adjacent motif (PAM); and determine how the complex tolerates mismatches between crRNA and the protospacer target. Noteworthy insights include the molecular details concerning PAM recognition by Cas8 and Cas5, crRNA-TS binding through the Cas7 backbone, and non-template strand displacement by Cas8 and Cas7 that allows for DinG recruitment. Uniquely, structural differences between the ways of how type IV-A1 and type IV-A3 associate with DinG were also characterized.

Major comments:

1. Overall, the structural biology work in this manuscript is very convincing. However, a portion of the results appear to have already been observed in a previous type IV-A1 structure of *P. aeruginosa* (Cui et al. *Molecular Cell*, 2023, which the authors cite). The key novelty is the comparison of the type IV-A1 and type IV-A3 structures, and of how they associate with their respective DinGs. Here the paper would benefit from some biochemical and/or molecular biology approaches that validate and complement their structural predictions.

We agree with the reviewer that additional experiments could complement our structural observations regarding CRISPR-RNP-DinG interactions. We have now included an additional *in vivo* experiment that dissects the type IV-A3 DinG interaction interface in our *lacZ*-CRISPRi assay (Fig. 8b), complementing the experiments on type IV-A1 (Fig. 6D). Regarding the comparison of types IV-A1 and IV-A3, one of the more striking differences is the full enclosure of the TS and NTS by type IV-A3 Cas7. We now present *in vitro* EMSA data probing substitutions in the Cas7 thumb helices that mediate this interaction (Fig. 5d). Indeed, the substitutions lead to less efficient DNA binding, comparable to the binding strength observed for type IV-A1.

A. The authors hypothesize that the type IV-A1 and type IV-A3 might possess different DNA binding kinetics, given their distinct PAM recognition mechanisms. An EMSA or fluorescent polarization experiment can test this.

Thank you for this suggestion. We now present EMSAs to compare the DNA binding kinetics. The EMSAs revealed that the type IV-A3 complex binds DNA approximately ten-fold stronger than the type IV-A1 complex (Fig. 4E,F). The assays further revealed that type IV-A3 is more sensitive to PAM-distal mismatches, suggesting different DNA-interaction modes.

B. Unlike type IV-A1, type IV-A3 DinG contacts the palm domains of Cas5, Cas7.1 and Cas7.2, in addition to interacting with the Cas7 WL motifs. Can these interactions be validated by non-structural means as well? For example, by mutating some of these interfaces, would type IV-A3 immunity against lambda-vir get reduced?

Yes. We now provide validation using our *lacZ*-CRISPRi assay, complementing the experiments on type IV-A1 (Fig. 6d). The assay revealed that the main determinant of binding is the interaction between the type IV-A3 DinG HD2 domain and the palm domains of Cas7 (Fig. 8b).

C. Does the translocation of DinG differ between A1 and A3?

Based on the structural resemblances (domain composition) and functional similarity, it seems likely that the ATP-dependent translocation mechanism along DNA is comparable between types A1 and A3. Notably, Cui *et al.* (2023, *Mol. Cell*) suggested that DinG may be capable of translocating along the CRISPR-RNP complex via the repetitive Cas7 interface. However, this observation was made using a non-native, nicked DNA substrate that may not reflect the structural constraints of a native DNA target. In our structures of DinG recruited to two divergent complexes bound to intact DNA, we do not observe any states corresponding to secondary DinG binding sites.

2. The authors should address why some experiments were only performed with one version of type IV-A and not the other. For example, mismatch tolerance was only done with the A1 system, while anti-phage defense was only performed with the A3 system. Consistent application of the assays would give a better comparison between the two systems.

We now provide complementary experiments in our *in vivo lacZ*-CRISPRi assay that dissect the type IV-A3-DinG interaction (Fig. 8b), which was previously only employed for type IV-A1 (Fig. 6d). Moreover, we have now added EMSA assays to compare the DNA binding strength and mismatch fidelity between both systems *in vitro* (Fig. 4e,f).

3. The title of the manuscript reads a bit too general, considering that there is already previous structural work on type IV-A. A better title would focus on the differences between the A1 and A3 system.

We have now changed the title of our manuscript to 'Structural Variation of Types IV-A1 and IV-A3-Mediated CRISPR Interference.'

Minor comments:

1. Lines 225-226: "This demonstrates that type IV-A3 functions robustly for PAM recognition." It is not clear what that statement means. Does it mean that the complex can recognize the PAM by redundant interactions?

We attempted to convey that mutating single amino acid residues does not affect the interference activity, where 'robust' simply means 'strong and unlikely to fail'. This observation has implications when considering the hypothetical presence of anti-CRISPR (Acr) proteins that may target the PAM-interaction site. It is conceivable that a degree of evolutionary plasticity could allow the evolution of type IV-A Acr-escaper variants. We have removed the sentence and now present this speculation in the discussion section.

2. Line 294-295: "...only mismatches in position 1, 4 and the combination of 2, 3 and 4 resulted in increased EOTs (Figure 4C)" This sentence is referring to the right panel of figure 4C; however, it is not representative of the figure being referenced. Mismatches in position 1, 4, and 9 results in increased EOTs; additionally, the combination 2, 3 and 4 is not shown in the graph.

Thank you. We have now divided panel 4C in 4C and 4D to make the data presentation more accessible. We also noticed that the combination of 2, 3 and 4 was mislabelled and corrected the graph label.

3. Line 300-304 would benefit from including the reference to the figure they are referring to, specifically citing Figure 4C.

We now refer to the Figure.

4. Line 303-305 "Mismatches in position 21-24 might be tolerated as they allow for completion of the R-loop structure, licensing DinG recruitment and thus interference." Is interesting to note that mismatches in 21-24 and 29-32 are tolerated. The authors argue that 21-24 is tolerated because it still allows for R-loop formation, while 29-32 do not bind to the other strand (shown by the structure). However, do the authors have insights on why 25-28 are not tolerated? The structure found that these nucleotides do bind to the complementary strand, however, if 21-24 is tolerated because it still allows for R-loop formation, it would seem that 25-28 also would allow for R-loop formation, and hence tolerated.

We speculate that mismatches in base pairs 25-28 prevent the completion of the R-loop structure in a conformation compatible with loading the NTS in DinG. We now include EMSA data showing that DNA mismatched in base pairs 25-28 is bound *in vitro* at a level comparable to the no-mismatch control (Fig. 4e). This suggests that the mismatches do not impair R-loop formation, but may impair the downstream interference regulation. Conceivably, DinG might not be able to become recruited, or activated, *in vivo*.

5. Line 364: "Figure 5: Cas8 and Cas5 guide the NTS." The figure title appears misleading, since the text argues that it is Cas8 and Cas7 that guide the NTS.

Thank you for spotting this typo. We have corrected the title of the Figure.

6. Figure 2E: Using the shield symbol to denote full CRISPR protection is misleading, as there is clear evidence of turbid plaque formation in some of the mutants (e.g. for amino acid substitutions K79A, Y51F, T114A, K119A). A better way to represent turbid plaque data (that lack obvious countable PFUs) is to set the limit of detection to the phage concentration where there are no visible plaques.

We have removed the shields to prevent confusion and set an LOD as suggested.

Reviewer #2 (Remarks to the Author)

I appreciate the effort of the authors to address my queries, I think they have made a good job. While some aspects of the mechanisms are still unclear I think the new version very has improves substantially.

Therefore I have no further comments and recommend the paper for publication

We thank Reviewer #2 for assessing our revised manuscript and for their publication recommendation.

Reviewer #3 (Remarks to the Author)

The authors address all of my comments in their revisions, and in my opinion, the manuscript is ready to be published.

We thank Reviewer #3 for their positive evaluation.